# Not All Documents Are What You Need for Extracting Instruction Tuning Data

Chi Zhang[*‡1,3], Huaping Zhong[*2], Hongtao Li[2], Chengliang Chai[†1], Jiawei Hong[2], Yuping Wang[1], Yuhao Deng[1], Jiacheng Wang[1], Yizhou Yan[5], Jiantao Qiu[3], Conghui He[†3], and Lei Cao[4]

[1]Beijing Institute of Technology
[2]SenseTime Research
[3]Shanghai Artificial Intelligence Laboratory
[4]University of Arizona
[5]Meta
zc315@bit.edu.cn , zhonghuaping@sensetime.com

## Abstract

Instruction tuning improves the LLMs performance but depends on high-quality training data. Recently, LLMs have been used to synthesize data, enhancing training with seeds like question-answer (QA) pairs. However, this synthesis often results in instruction examples similar to the seeds, lacking diversity and biasing real applications. Thus, we propose to extract instruction tuning data from web corpus with much rich knowledge. The most straightforward strategy is to quickly retrieve domain specific documents from the corpus and then extract all QA pairs of these documents for tuning LLMs, which has two main limitations. (1) Extracting all QA pairs using LLMs is prohibitively expensive; and (2) These extracted pairs are not all beneficial for the downstream applications, and incorporating all of them for tuning may even hurt the model performance. To overcome the limitations, we introduce EQUAL, an **E**ffective and scalable data extraction framework that iteratively interleaves document selection and extract high-**QUAL**ity QA pairs to optimize instruction tuning. EQUAL first clusters the document set based on the embeddings generated by contrastive learning. Then it leverages the multi-armed bandit based strategy to quickly identify document clusters where can extract high-quality QA pairs for training. This iterative framework significantly reduces computational costs while improving model performance much. Experiments on AutoMathText, KnowledgePile and StackOverflow across 13 downstream tasks demonstrate that EQUAL reduces computational costs by 5–10× while improving accuracy by 2.5% on LLaMA-3.1-8B, Qwen2.5-7B and Mistral-7B. Code and data is available at https://anonymous.4open.science/r/EQUAL-DD20.

## 1 Introduction

Previous studies have shown that instruction tuning enables the powerful reasoning capability of Large Language Models (LLMs) (Ouyang et al., 2022; Achiam et al., 2023; Dubey et al., 2024), but requires sufficient high-quality training data (Ntoutsi et al., 2020; Yu et al., 2023; Shah et al., 2024). However, although the weights of the open LLMs are publicly available, the datasets employed to fine-tune these models are generally private. This lack of data accessibility limits the opportunities to effectively adapt LLMs to targeted domains (Cobbe et al., 2021b; Hendrycks et al., 2021).

Recently, leveraging LLMs to synthesize instruction tuning data (Li et al., 2024a; Yue et al., 2024; Luo et al., 2023; Yu et al., 2023; Li et al., 2024a; Ding et al., 2024) has attracted much attention as an effective solution to enrich the original training data based on some seeds (*e.g.*, original

---

[*]Equal Contribution.
[†]Correspondence to: {ccl@bit.edu.cn, heconghui@pjlab.org.cn}
[‡]This work was done during the internship at Shanghai Artificial Intelligence Laboratory.

question-answer pairs, knowledge bases, etc.), thanks to the powerful understanding and generative capabilities of LLMs. However, achieving high-quality synthetic instruction data is challenging because LLM-based generation tends to closely imitate seed examples. When those seeds lack diversity, the synthesized data inherits this shortcoming, leading to degraded overall quality (Guo et al., 2024b; Li et al., 2024c; Xu et al., 2024; Ding et al., 2024).

**Data Extraction from Documents.** In reality, there is plenty of high-quality web corpus (*e.g.*, Common Crawl) which contains rich knowledge and can be leveraged as high-quality instruction data. However, this wealth of knowledge is widely spread within the corpus. Recently, (Yue et al., 2024) proposed a method to retrieve domain-specific documents from a large web corpus, followed by employing high-performance LLMs to extract QA pairs from these documents and then using the extracted QA pairs to fine-tune an LLM. However, it has the following limitations.

*Prohibitive Computational Cost.* To extract a high-quality instruction tuning dataset, it uses LLMs to repeatedly scan and analyze all the documents to extract question-answer (QA) pairs, each of which often requires multiple LLM calls (Gilardi et al., 2023; Yue et al., 2024). Consequently, this process is prohibitively expensive, especially when there are a large number of documents to process. Solving this problem requires largely reducing the number of candidate documents, *e.g.*, by discovering the documents most valuable to instruction tuning.

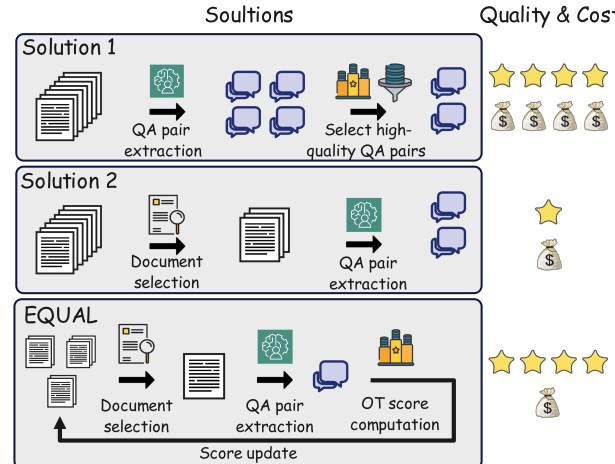

Figure 1: Number of training iterations on different models

*Instruction QA Pairs.* Even if an organization could afford extracting all domain-specific QA pairs from a large number of documents, blindly incorporating all of them to fine-tune an LLM could potentially degrade the model performance due to the presence of a significant amount of low quality data. Specifically, a large corpus inevitably includes wild and noisy data with heterogeneous distribution that can degrade model performance on downstream tasks. Therefore, it is necessary to judiciously identify high-quality QA pairs for extraction.

**Intuitive Solutions.** There are two intuitive solutions to address the above limitations. Solution ①: extract all QA pairs first and then select high-quality pairs. Solution ②: Select high-quality documents that potentially contain high-quality pairs first and then extract QA pairs from them. Unfortunately, neither of the above two methods can solve both limitations. To be specific, for solution ①, although it can achieve good model performance, extracting all pairs beforehand is still costly. Solution ② is cost-effective, but it is difficult to accurately discover high-quality documents because documents and pairs have different feature distributions. Note that in this case, the quality of these documents should be measured by the potential contribution of the QA pairs to the target distribution, which is not aligned with the data quality of the original documents, e.g., dirty data, duplication. In other words, even if the embedding of a QA pair is close to that of a document, it does not necessarily indicate that the QA pair is close to the pair potentially extracted from the document.

**Key Idea.** To address both limitations, our key idea is to interleave document selection and QA pairs extraction. During this iterative process, the extracted QA pairs help capture the relationship between the document and the pair distribution more and more accurately, and at the same time, the selected documents improve consistently.

**Our Proposal.** Inspired by the above idea, we propose EQUAL, a scalable and effective data extraction framework for constructing QA pairs from documents, aiming to enhance the LLMs instruction tuning. To be specific, EQUAL first clusters over the heterogeneous document set considering the feature similarities of QA pairs extracted from these documents. To achieve this, we introduce a warm-up step using contrastive learning to align the feature space between documents and QA pairs. In this way, EQUAL effectively identifies those high-quality clusters by sampling and extracting QA pairs from them to save cost. Afterwards, we propose a Multi-arm Bandit (MAB) based technique

to iteratively select the clusters. As the reward function, it predicts the benefit of QA pairs that potentially could be extracted from the clusters. More specifically, in each iteration, `EQUAL` tends to select the cluster where documents can produce QA pairs that are likely to benefit the target model performance. This benefit is measured by the *optimal transport (OT) score*, where a higher benefit score indicates a smaller difference between the distributions of the QA pairs in the cluster and the target distribution. Then, given the selected cluster, `EQUAL` samples some documents from it, extracts QA pairs using LLMs, and in turn updates the optimal transport score of this cluster accordingly. In this iterative process, we precisely estimate the distribution of the QA pairs in a document cluster without having to conduct extraction over all documents.

Moreover, leveraging the upper confidence bound technique in MAB, `EQUAL` promotes the potentially low-quality, thus under-sampled clusters. Therefore, it improves the diversity of the extraction data. This balance between exploration and exploitation effectively avoids reaching a local optimum.

To summarize, we make the following contributions:

(1) We propose `EQUAL`, a novel framework for data extraction from documents to enhance LLMs instruction tuning with high scalability.

(2) We incorporate an iterative MAB solution to first cluster the documents and extract data from these clusters, achieving a good exploration-exploitation trade-off. We also propose a warm-up strategy to align the features of documents and QA pairs.

(4) Extensive experiments on datasets (AutoMathText, KnowledgePile and StackOverflow) with more than 1 million documents and 13 popular downstream tasks demonstrate that `EQUAL` significantly outperforms baseline methods by saving 5-10$\times$ computation resources consumption while still improving 2.5% in accuracy (train/test on Llama-3.1-8B, Qwen2.5-7B and Mistral-7B model).

## 2 PRELIMINARY

We first introduce the necessity of extracting and selecting QA pairs for instruction tuning, followed by our problem definition.

**Instruction tuning** (a.k.a., supervised fine-tuning (SFT)) aims to adapt a base LLM to specific domains or user needs, thereby enabling better performance on downstream tasks (Wang et al., 2024b; 2023a; Zhang et al., 2023). Consequently, many studies on data selection for SFT (Li et al., 2023b; Xia et al., 2024; Ni et al., 2024) emphasize selecting domain-beneficial data from a candidate pool with reference to the target capabilities. This is because fine-tuning LLMs without careful data selection—for instance, using QA pairs extracted indiscriminately from all documents in the candidate set—can impede the model's ability to acquire target capabilities. Empirical results from prior works (Kim et al., 2023; Muennighoff et al., 2022; Wang et al., 2023b) demonstrate that LLMs fine-tuned on targeted subsets of data outperform those trained on the full SFT dataset, underscoring the importance of relevance-aware data selection.

**QA Pair Extraction.** To construct SFT data, given a large number of documents, people always leverage advanced LLMs to extract and refine QA pairs. More specifically, LLMs such as Qwen2.5-72B (Yang et al., 2024a) are utilized to extract QA pairs within documents. By incorporating examples within the prompt, we guide the model to focus on the desired QA pairs while filtering out markups, boilerplate, and other irrelevant content. Although Qwen2.5-72B demonstrates impressive capabilities, the extracted QA pairs still exhibit issues such as improper formatting, missing answers, or mismatched responses (Li et al., 2023a; Honovich et al., 2022; Chen et al., 2023). To address these problems, we employ further refinement to these QA pairs using Qwen2.5-72B, which has demonstrated significant enhancement in the quality of the extracted pairs (Xu et al., 2023). The prompts for extraction and refinement are provided in Appendix W.

Overall, the entire extraction process requires each entire document as input and repeatedly calling high-performance LLMs, which is rather expensive. Thus, in this paper, we focus on reducing the number of documents to be extracted to save the cost while keeping high model accuracy. However, how to improve the quality of each extracted pair within each document is orthogonal to this work.

**Problem Definition.** Formally, we study the problem of data extraction from a candidate document pool $\mathcal{D}_c$ to extract QA pairs for instruction tuning. Formally, given $\mathcal{D}_c$ and a reference set $\mathcal{D}_r$ for

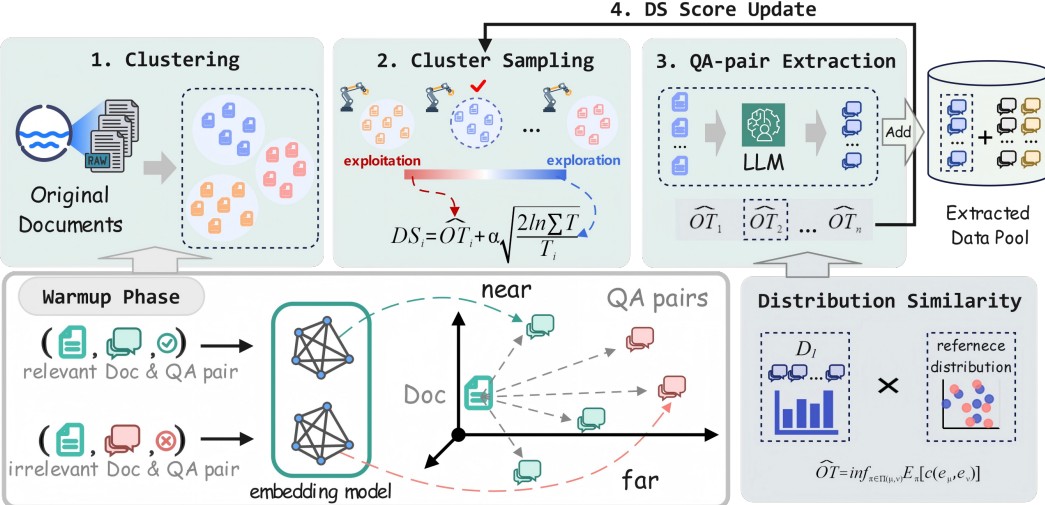

Figure 2: The Overall Framework of `EQUAL`.

instruction tuning, the problem is to select a subset $\mathcal{D}_b \subset \mathcal{D}_c$ from which a set of QA pairs $\mathcal{Q}$ are extracted to fine-tune an LLM $M$, aiming to minimize the loss of the updated model $M'$ on $\mathcal{D}_r$.

# 3 PROPOSED APPROACH.

## 3.1 THE EQUAL FRAMEWORK

**Multi-Armed Bandit (MAB)** (Vermorel & Mohri, 2005) is an effective framework that makes decisions over time under uncertainty. This consists of $N$ possible actions, each known as an $arm$. Pulling an arm indicates sampling from this arm to capture its reward distribution more accurately. This framework characterizes an agent that iteratively gains new knowledge by pulling arms that are rarely visited (*i.e.*, exploration) while using current knowledge to enhance its decisions by pulling arms already with a high reward (*i.e.*, exploitation). The agent aims to balance the exploration and exploitation to maximize their overall reward throughout the given time span.

**Bridging EQUAL and MAB.** The overall process of `EQUAL` is illustrated in Fig 2 and Algorithm 1. To reduce the computational cost, we first cluster all documents in the candidate dataset $\mathcal{D}_c$ (line 1) such that the QA pairs extracted from each cluster are similar (Step-1 in Fig 2, see § 3.2 for details). Thus, we can neglect these low-quality clusters to save the cost. To precisely measure the quality of the cluster $C_i$, the most straightforward way is to extract all QA pairs and compare their distribution with that of the reference data, measured by the optimal transport score (see § 3.3 for details), denoted by $OT_i$, but it is still very expensive. Hence, we iteratively sample from these clusters to estimate the score. Our key idea is inspired by a natural connection between `EQUAL` and MAB.

At a high level, each cluster can be regarded as an arm of MAB. `EQUAL` iteratively selects a cluster, samples some documents and extract pairs from it (*i.e.*, pulling an arm). To be specific, as shown in Step-2 of Figure 2, a cluster with a high estimated optimal transport score ($\hat{OT}_i$) tends to be selected. The higher the score, the smaller difference between the distributions of QA pairs from this cluster and the reference data. Moreover, clusters that are rarely visited (denoted by the sampling frequency $T(C_i)$) tend to be selected as well to explore more diverse documents. Overall, putting $\hat{OT}_i$ and $T(C_i)$ together, we use the document sampling (DS) score to measure the quality of a cluster, which can achieve a good exploration-exploitation trade-off. Subsequently, as shown in Step-3, we obtain these extracted pairs (line 5-7) and update the score $\hat{OT}_i$ of the corresponding cluster (line 8). As more pairs from cluster $C_i$ are extracted, the estimated score $\hat{OT}_i$ will become more accurate.

Next, we illustrate the details of the `EQUAL` framework.

**DS Score Computation.** Following the upper confidence bound (Auer, 2002) in the typical MAB framework, we define the DS score $DS_j$ of the cluster $C_j$ to effectively balance the exploration (*i.e.*,

data diversity) and exploitation (*i.e.*, data quality) as follows.

$$DS_j = \hat{OT}_j + \alpha\sqrt{\frac{2\ln\sum_{C_k\in\mathcal{C}}T(C_k)}{T(C_j)}} \tag{1}$$

where $T(C_j)$ denotes the frequency of documents sampled from cluster $C_j$, $\sum_{C_k\in\mathcal{C}}T(C_k)$ denotes the total sampling times from all clusters. $\alpha$ is set as $\frac{1}{\sum_{C_k\in\mathcal{C}}T(C_k)+1}$ (Hao et al., 2019), which provides higher weight to exploration in early stages, but in later stages, it provides higher weight to exploitation (line 9-11).

**Update the DS Score.** In each iteration, a subset of documents $B_i$ is sampled from the selected cluster $C_i$ with a high DS score, and a set of QA pairs $Q_i$ is then extracted from the documents in $B_i$. The OT score of $C_i$ will be updated as follows.

$$\hat{OT}_i = \mathcal{OT}(\cup Q_i, \mathcal{D}_r), \quad T(C_i)+ = 1 \tag{2}$$

where $\cup Q_i$ denotes all the extracted QA pairs in cluster $C_i$ obtained from the beginning and $\mathcal{OT}(\cdot)$ denotes the function of computing the OT score. Then we update the DS score of all clusters.

**Extracted Pairs Collection.** As shown in Figure 2, in each iteration, we add the extracted QA pairs $Q_i$ to the extracted data pool $\mathcal{D}_e$. Finally, we use the pairs in the pool to fine-tune the LLMs.

---

**Algorithm 1:** EQUAL Algorithm

**Input:** Candidate data pool $\mathcal{D}_c$, reference set $\mathcal{D}_r$, extracted data ratio $\gamma$.

**Output:** Extracted data pool $\mathcal{D}_e$.

1   $\mathcal{C} = \text{Cluster}(\mathcal{D}_c)$;
2   $\mathcal{D}_e = \emptyset$;
3   **while** $|\mathcal{D}_e| < \gamma|\mathcal{D}_c|$ **do**
4      Select cluster $C_i$ with the highest DS Score;
5      Sample $B_i$ documents from $C_i$;
6      Extract QA pairs $Q_i$ from $B_i$;
7      $\mathcal{D}_e = \mathcal{D}_e \cup Q_i$;
8      $\hat{OT}_i = \mathcal{OT}(\cup Q_i, \mathcal{D}_r)$, $T(C_i)$ += 1;
9      **for** $C_j$ *in* $\mathcal{C}$ **do**
10         $DS_j = \hat{OT}_j + \alpha\sqrt{\frac{2\ln\sum_{C_k\in\mathcal{C}}T(C_k)}{T(C_j)}}$;
11      **end**
12 **end**
13 **Return** $\mathcal{D}_e$;

---

## 3.2   Warm-up for Clustering

**Motivation.** Considering that the original documents contain much irrelevant content with downstream applications, there exists a discrepancy between the feature space of document embeddings and that of the QA pairs extracted from them. Obviously, we hope that similar QA pairs fall into the same cluster, but similar documents do not necessarily indicate similar pairs if we cluster purely based on feature embeddings of documents. However, it is rather expensive to extract all pairs and then cluster. Therefore, to improve the clustering quality, we propose to incorporate a warm-up step to align the two feature spaces using contrastive learning (Khosla et al., 2020).

**Feature Alignment.** In the warm-up stage, we first randomly sample a small proportion of the documents from the candidate data pool and use LLMs to extract QA pairs. Then, we fine-tune the model (*i.e.*, BAAI/bge-en-v1.5) used for document embedding to capture the deep connection between the original documents and the extracted QA pairs. Specifically, we treat the sampled documents and these extracted QA pairs as positive training examples (denoted by $(d, q^+)$) for contrastive learning. To generate negative examples $q^-$ for a document $d$, we conduct negative sampling from all current QA pairs. Then we train with the following loss function:

$$L = -log\frac{e^{sim(d,q^+)}}{e^{sim(d,q^+)} + \sum e^{sim(d,q^-)}} \tag{3}$$

where $sim$ denotes the cosine similarity between the embedding of a document $d$ and QA pair $q$. In this way, documents containing similar QA pairs tend to be closer in the embedding space and are thereby grouped together in the same cluster.

## 3.3   Measuring Cluster Benefits

Heuristic methods (Xia et al., 2024; Xie et al., 2023) estimate the pointwise benefit of each data point (i.e., its contribution to the target capabilities) and simply aggregate these benefits as cluster benefits,

implicitly assuming that each data point contributes independently of the others. However, this assumption fails even in simple linear regression tasks, as systematically demonstrated in (Hu et al., 2024). In contrast, `EQUAL` formulates targeted data selection as a distribution matching problem, aiming to identify a subset of candidate data that closely matches the target distribution. Specifically, `EQUAL` computing the distribution similarity of extracted data and reference data utilizing Optimal Transport (OT) (Villani et al., 2009), which is widely adopted to compute the minimal cost of transforming one distribution into another (more details in Appendix H). Specifically, the lower the transportation cost, the closer the two distributions, indicating that the extracted data is more beneficial for the target distribution.

In our scenario, suppose that the distribution of extraction data and reference set is $\mu$ and $\nu$ separately. The transportation cost from $\mu$ to $\nu$ can be calculated by $\mathcal{OT}(\mu, \nu)$:

$$\mathcal{OT}(\mu, \nu) \overset{\text{def}}{=} \inf_{\pi \in \Pi(\mu, \nu)} \mathbb{E}_{(e_\mu, e_\nu) \sim \pi} [c(e_\mu, e_\nu)] \tag{4}$$

where $e_\mu$ and $e_\nu$ denote the embedding of extracted QA pairs $q_\mu, q_\nu$ from the two distributions, $\Pi(\mu, \nu)$ denotes the set of all joint distributions $\pi(e_\mu, e_\nu)$ with marginals $\mu(e_\mu)$ and $\nu(e_\nu)$. Here, $c(e_\mu, e_\nu) : \mathbb{X} \times \mathbb{X} \to \mathbb{R}$ is the cost function for moving $e_\mu$ to $e_\nu$, where $\mathbb{X}$ denotes the entire embedding space in `EQUAL`. To be specific, we use $1 - \frac{e_\mu^T e_\nu}{\|e_\mu\| \|e_\nu\|}$ as the transportation cost between $e_\mu, e_\nu$, which is a popular choice to measure the semantic dissimilarity (Pennington et al., 2014). Then given two distributions $\mu$ and $\nu$, there are numerous possible mappings (i.e., $\pi \in \Pi$) between pairs from these distributions. The cost for each mapping can be calculated by various $c(e_\mu, e_\nu)$ within the mapping, and the OT score represents the minimum cost among all the mappings.

## 4 EXPERIMENT

In this section, we fine-tune the base models in different domains and conduct sufficient ablation studies to demonstrate the efficiency and effectiveness of `EQUAL`. More experiments such as enlarging the model size and the diversity of downstream tasks is provided in the Appendix I.

### 4.1 EXPERIMENT SETUP

**Training Settings.** We evaluate `EQUAL` using two foundational models (i.e., LLAMA-3-8B, Qwen2.5-7B and Mistral-7B) and two training settings, i.e., full fine-tuning (FULL) and Low-Rank Adaption (LoRA). In both training scenarios, the batch size is set to 512 and the maximum learning rate is set as $1 \times 10^{-5}$ with a cosine decay schedule. For the FULL setting we train the extracted data for 2 epochs on 32 H100 GPUs, while for the LoRA setting we train the extracted data for 4 epochs on 16 H100 GPUs. For the warm-up stage, we randomly select 5% documents from $\mathcal{D}_c$ to extract QA pairs and then use contrastive learning to fine-tune the original embedding model `BAAI/bge-en-v1.5 model`, and then it is employed to generate document embeddings for subsequent clustering.

**Dataset Preparation.** In our evaluation, we use AutoMathText (Zhang et al., 2024), KnowledgePile (Fei et al., 2024) (Appendix I and M) and StackOverflow (created by us) datasets as the candidate data pool $\mathcal{D}_c$ for mathematical, general and coding tasks respectively. AutoMathText totally contains 4.9M documents, from which we select 1.4M ones by filtering documents with a metadata score of $lm\_q_1q_2\_score < 0.5^*$ to exclude math-irrelevant content. We use the dataset after filtering as the candidate data pool. StackOverflow is crawled by us from `stackoverflow.com`, which contains 1.2M documents in total. Then we implement an $n$-gram filtering (Guo et al., 2024a) to ensure that our training data is not contaminated by information from the downstream tasks. For the reference (validation) set $\mathcal{D}_r$, we respectively use the training set of GSM8K (Cobbe et al., 2021a) and MBPP (Austin et al., 2021) for math and code domains, which are both widely used language modeling tasks and often serve as a validation benchmark for instruction tuning. During the clustering process, documents from $\mathcal{D}_c$ are clustered into 1,000 clusters using the $k$-means algorithm. The number of clusters is automatically determined by the Elbow (Syakur et al., 2018) method in `EQUAL`.

---

$^*lm\_q_1q_2\_score$ is a metadata attribute for each document in AutoMathText ranging from [0, 1], which quantifies the document's relevance, quality, and educational value in the context of mathematical intelligence.

Table 1: Comparison with other algorithms in test accuracy (%) on AutoMathText and StackOverflow. The best results are highlighted. We run each experiment for three times and report the average.

| Model | | LLAMA-3-8B | | | | | | Mistral-7B | | | | | |
|---|---|---|---|---|---|---|---|---|---|---|---|---|---|
| Domain | | Math | | | Code | | | Math | | | Code | | |
| Methods | | GSM8K | MATH | FLOPs | HUMANEVAL | MBPP | FLOPs | GSM8K | MATH | FLOPs | HUMANEVAL | MBPP | FLOPs |
| Base Model | | 55.19 | 23.04 | - | 31.1 | 51.9 | - | 45.10 | 14.80 | - | 23.2 | 41.8 | - |
| Random | LoRA | 63.76 | 30.26 | 8.05 | 31.1 | 53.7 | 6.32 | 54.21 | 20.78 | 7.96 | 28.0 | 45.0 | 6.06 |
| Avg-sim | LoRA | 65.64 | 30.12 | 114.79 | 31.7 | 52.6 | 65.53 | 51.33 | 21.86 | 113.49 | 28.1 | 45.1 | 65.45 |
| Perplexity | LoRA | 63.61 | 30.94 | 134.35 | 31.1 | 54.6 | 71.07 | 51.40 | 22.18 | 134.18 | 25.6 | 44.4 | 70.9 |
| Influence | LoRA | 63.46 | 28.10 | 240.31 | 33.5 | 55.0 | 130.46 | 53.45 | 18.62 | 236.59 | 26.2 | 41.5 | 129.68 |
| LLM-scoring | LoRA | 65.11 | 30.55 | 271.56 | 34.3 | 53.5 | 141.38 | 53.02 | 19.33 | 266.51 | 27.7 | 44.6 | 137.37 |
| Rewriting | LoRA | 62.21 | 27.05 | 18.11 | 30.7 | 54.1 | 13.15 | 50.19 | 17.72 | 17.90 | 26.3 | 44.3 | 13.01 |
| Perplexity-MAB | LoRA | 64.52 | 30.48 | 18.18 | 34.6 | 55.0 | 13.24 | 51.78 | 22.10 | 17.36 | 27.4 | 45.6 | 12.99 |
| Influence-MAB | LoRA | 65.73 | 30.78 | 23.37 | 35.3 | 55.1 | 19.13 | 55.12 | 19.58 | 18.44 | 28.7 | 44.7 | 18.18 |
| EQUAL(Ours) | LoRA | **67.32** | **31.86** | 17.75 | **36.0** | **55.3** | 12.99 | **57.54** | **23.56** | 17.57 | **31.3** | **46.7** | 12.64 |
| Random | FULL | 68.92 | 32.46 | 8.83 | 42.7 | 52.3 | 7.19 | 61.41 | 25.76 | 8.74 | 31.1 | 44.2 | 6.84 |
| Avg-sim | FULL | 70.35 | 33.18 | 115.66 | 46.1 | 53.2 | 66.31 | 61.88 | 26.16 | 114.44 | 37.8 | 45.1 | 63.20 |
| Perplexity | FULL | 64.52 | 33.56 | 150.28 | 44.5 | 50.5 | 71.85 | 55.04 | 27.38 | 148.03 | 32.6 | 44.7 | 71.07 |
| Influence | FULL | 65.20 | 29.64 | 256.94 | 39.6 | 53.7 | 131.24 | 56.18 | 22.08 | 248.97 | 35.6 | 45.6 | 129.51 |
| LLM-scoring | FULL | 68.38 | 33.19 | 273.91 | 46.9 | 53.7 | 142.73 | 56.19 | 22.72 | 17.90 | 34.3 | 46.1 | 139.16 |
| Rewriting | FULL | 64.47 | 30.62 | 18.71 | 43.4 | 50.6 | 13.78 | 57.21 | 22.67 | 18.33 | 33.3 | 43.6 | 13.67 |
| Perplexity-MAB | FULL | 65.28 | 32.92 | 18.96 | 44.5 | 49.1 | 13.76 | 56.44 | 24.20 | 18.44 | 34.1 | 42.9 | 13.42 |
| Influence-MAB | FULL | 67.78 | 32.86 | 25.62 | 46.3 | 53.5 | 19.91 | 60.42 | 24.44 | 19.30 | 34.8 | 44.0 | 18.79 |
| EQUAL(Ours) | FULL | **73.01** | **35.10** | 18.55 | **49.4** | **56.3** | 13.50 | **67.73** | **28.28** | 18.18 | **39.1** | **50.6** | 13.07 |

**Baselines.** We compare EQUAL with several baselines. (1) Random. We randomly sample documents to extract QA pairs from $\mathcal{D}_c$ for fine-tuning. (2) All(Mammoth). We fine-tune our model using the QA pairs extracted from all the documents in candidate data pool $\mathcal{D}_c$, which is the same method used in Mammoth (Yue et al., 2024). (3) Rewriting (Yu et al., 2023) synthesize new QA pairs based on existing QA pairs (specifically, the reference set $\mathcal{D}_r$ in our setting) using LLM. We synthesize the same number of pairs as other baselines. (4) Avg-sim. We extract QA pairs from all documents in $\mathcal{D}_c$. Then we select QA pairs with the highest average similarities with the ones in $\mathcal{D}_r$. For each QA pair, we compute the embedding similarities between the pair and all pairs in $\mathcal{D}_r$, and compute the average. (5) Perplexity (Li et al., 2024b) extract QA pairs from all documents in $\mathcal{D}_c$. Then we select QA pairs with the highest perplexity scores. (6) Influence (Xia et al., 2024) extract QA pairs from all documents in $\mathcal{D}_c$. Then we select extracted QA pairs with the highest influence scores. (7) LLM-scoring (Wettig et al., 2024) employs high-performing LLMs to evaluate the quality score of documents. (8) Perplexity-MAB utilizes perplexity (Li et al., 2024b) score as the reward of MAB to select documents for extracting QA pairs. (9) Influence-MAB utilizes influence (Xia et al., 2024) score as the reward to select documents. (10) EQUAL is our full-fledged solution.

**Metric.** We evaluate the quality of extracted data by accessing the LLM performance, which has been fine-tuned with these data on several commonly used downstream tasks. (1) Math domain: GSM8K (Cobbe et al., 2021a) and MATH (Hendrycks et al., 2021) are utilized for evaluating math tasks. (2) Code domain: the fine-tuned LLM is evaluated on HUMANEVAL (Chen et al., 2021) and MBPP (Austin et al., 2021) datasets. We also report FLOPs to quantify the total GPU cost across the following three stages: data extraction, data selection, and model training, with details in Appendix F.

## 4.2 RESULT

**Overall Performance.** In Table 1, we acquire 5% documents (70k from AutoMathText and 60k from StackOverflow) for each baseline. We can observe that EQUAL surpasses all the baseline methods on accuracy across all models and downstream tasks. Specifically, when implementing Full fine-tuning on Llama-3.1-8B, EQUAL achieves an accuracy improvement of 4.09% on GSM8k and 2.64% on MATH compared with Influence, while saving approximately $5\times$ w.r.t. the computational cost. EQUAL surpasses Rewriting due to the fact that the QA pairs generated by Rewriting are quite similar to those QA pairs in $\mathcal{D}_r$, resulting in limited data diversity. Besides, the pairs directly generated by LLMs might be error-prone due to the hallucinations. EQUAL outperforms Perplexity and Perplexity-MAB because perplexity score is solely based on the inherent complexity of the QA pairs to select extracted data without considering the downstream tasks. Besides, EQUAL outperforms Influence and Influence-MAB because influence function is easily affected by the length of the sequence (Xia et al., 2024), often leading to the selection of pairs with fewer tokens (more details in Appendix G). Also, EQUAL outperforms LLM-score and Avg-sim because it selects data based on the similarities between QA pairs, without considering the overall distribution. In terms of the computational cost, we can observe that the FLOPs consumed by the Avg-sim, Influence and Perplexity extraction method are notably high. This is due to their necessity of extracting

Table 2: Comparison with `Random` at different ratios.

| Method | LoRA | | | | | | FULL | | | | | |
|---|---|---|---|---|---|---|---|---|---|---|---|---|
| | Math | | | Code | | | Math | | | Code | | |
| | GSM8K | MATH | FLOPs | HUMANEVAL | MBPP | FLOPs | GSM8K | MATH | FLOPs | HUMANEVAL | MBPP | FLOPs |
| Random (5%) | 63.76 | 30.26 | 8.05 | 31.1 | 53.7 | 6.32 | 67.40 | 32.46 | 8.83 | 42.7 | 52.3 | 7.19 |
| Random (10%) | 66.03 | 30.82 | 16.37 | 32.3 | 53.8 | 13.11 | 68.92 | 34.54 | 17.50 | 43.6 | 54.6 | 14.33 |
| Random (20%) | 65.05 | 31.76 | 32.14 | 33.5 | 54.1 | 25.97 | 70.05 | 36.18 | 34.61 | 44.1 | 55.0 | 27.78 |
| All | 65.43 | 32.90 | 155.6 | 34.8 | 55.3 | 122.5 | 70.28 | 40.02 | 164.9 | 45.6 | 56.0 | 137.8 |
| EQUAL (5%) | 67.32 | 31.86 | 17.75 | 36.0 | 55.3 | 12.99 | 73.01 | 35.10 | 18.55 | 49.4 | 56.3 | 13.50 |
| EQUAL (10%) | 68.10 | 31.66 | 25.21 | 38.8 | 56.0 | 19.51 | 74.46 | 38.19 | 27.38 | 50.1 | 56.0 | 19.76 |
| EQUAL (20%) | 68.69 | 33.43 | 40.11 | 39.6 | 55.5 | 33.17 | 74.40 | 41.40 | 43.67 | 49.6 | 56.4 | 33.51 |

the QA pairs from all the documents in $\mathcal{D}_c$, which incurs prohibitive cost. Additionally, since the influence score used in `Influence` requires to compute gradients during back propagation, leading to higher FLOPs consumption than `EQUAL`. In contrast, it can be seen that for the influence score used in `Influence` and the perplexity score used in `Perplexity`, when combined with the MAB framework, comparable results can be achieved at a lower computational cost, which demonstrates the efficiency of the MAB strategy. The detailed performance of `EQUAL` on Qwen2.5-7B can be found in the Appendix S.

Table 2 presents a comparison between `EQUAL` and `Random` using various QA pair extraction ratios ranging from 5%, 10%, and 20% up to 100% (i.e., high-cost extraction from all documents). Interestingly, we find that the extracting of just 5% of QA pairs for most tasks produces superior results compared to the use of complete $\mathcal{D}_c$. This demonstrates the effectiveness of `EQUAL`. Even for the difficult task MATH, extracting QA pairs from only 20% to the documents in $\mathcal{D}_c$ can achieve comparable performance to `All(Mammoth)` across all the training settings. This is because not all the QA pairs extracted from all the documents in $\mathcal{D}_c$ might contribute to the target tasks.

## 4.3 ABLATION STUDY

In this section, we demonstrate the effectiveness of Warm-up phase, Multi-Armed Bandit (MAB) and Document Sampling (DS) score through experiments conducted with `no-warmup`, `no-MAB` and `no-DS` settings, which is shown in Table 3. Also, we conduct several ablation studies $w.r.t.$ the number of clusters, different clustering algorithms etc., and the results are illustrated in Figure 3.

**Effectiveness of Warm-up Phase.** As shown in Figure 3, points of the same color represent QA pairs extracted from the documents in the same cluster. Figure 3a shows the clustering results obtained by directly using the existing model to compute embeddings for each document, while Figure 3b presents the results after fine-tuning the embedding model during warm-up phase. Since the original documents contain many contents irrelevant to the extracted QA pairs, leveraging the document embeddings directly to cluster will lead to the inconsistencies of QA pair embeddings within each cluster. It can be observed that the warmup phase effectively aligns the embeddings of documents and their corresponding QA pairs. Also, we also conduct the experiment `no-warmup`, which use the original embedding model `BAAI/bge-en-v1.5` to generate document embeddings for clustering. Following this, similar to `EQUAL`, both the MAB framework and optimal transport are employed to extract QA pairs iteratively based on the clusters. Table 3 illustrates that `EQUAL` surpasses `no-warmup` across all experimental settings, demonstrating the effectiveness of the warm-up phase.

**Effectiveness of MAB.** In this section, we evaluate the performance of MAB by contrasting it with a simple method called `no-MAB`, which extracts QA pairs directly from several clusters with distributions similar to the reference set without iterative extraction like MAB. Specifically, we start by sampling a small set of documents from each cluster and then extract QA pairs. Subsequently, optimal transport is employed to assess the similarity in distribution between these pairs and the reference set, serv-

Table 3: Effectiveness of Warm-up phase, Multi-Armed Bandit, Document Sampling (DS) score in `EQUAL`.

| Domain | | Math | | Code | |
|---|---|---|---|---|---|
| Method | | GSM8K | MATH | HUMANEVAL | MBPP |
| no-warmup | LoRA | 64.05 | 30.82 | 32.8 | 54.1 |
| no-MAB | LoRA | 66.13 | 30.55 | 33.3 | 53.3 |
| no-DS | LoRA | 65.59 | 31.08 | 34.4 | 53.7 |
| EQUAL | LoRA | 67.32 | 31.86 | 36.0 | 55.3 |
| no-warmup | Full | 69.73 | 33.51 | 44.3 | 53.8 |
| no-MAB | Full | 71.90 | 33.25 | 46.9 | 55.5 |
| no-DS | Full | 70.77 | 33.40 | 47.6 | 54.6 |
| EQUAL | Full | 73.01 | 35.10 | 49.4 | 56.3 |

ing as the score for each cluster. Then, the documents within the top-scoring clusters are utilized

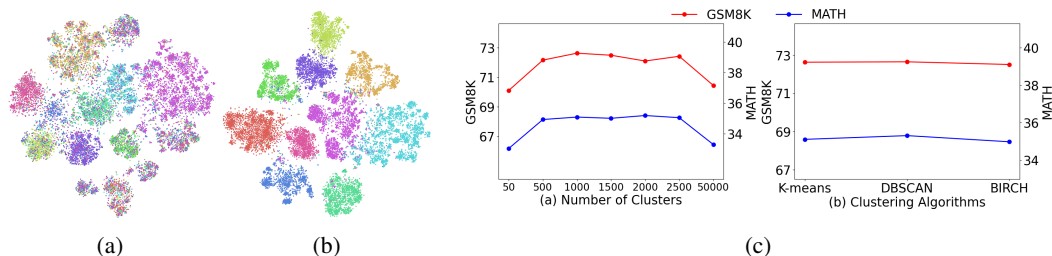

Figure 3: (a) shows the clusters based on the original embedding model; (b) shows the clusters based on the fine-tuned embedding model; (c) ablation study of cluster numbers and clustering algorithms.

to extract QA pairs and fine-tune the LLM. As illustrated in Table 3, `EQUAL` outperforms `no-MAB`, because during the data extraction process, the OT score for each cluster is updated dynamically based on the previously extracted QA pairs, thereby enhancing the accuracy of the subsequent selection.

**Effectiveness of Document Sampling (DS) score.** We evaluate the effectiveness of DS score by comparing `EQUAL` with `no-DS`, which utilizes the average similarities with QA pairs in $\mathcal{D}_r$ as the MAB reward. As shown in Table 3, `EQUAL` surpasses the `no-DS` across all the settings. This indicates that DS score provides a more precise estimation of the distributional similarity between the extracted data and the reference set, hence better aligning with downstream applications.

**Number of Clusters.** We use the Elbow (Syakur et al., 2018) method to identify the optimal cluster numbers for AutoMathText and StackOverFlow datasets. In Figure 3(a), we plot the accuracy of `EQUAL` with different cluster numbers. When the cluster number is around 1000 (the selected optimal number for both datasets), the model consistently performs well. However, a very small number of clusters (*i.e.*, $k = 50$) leads to poor accuracy (2.90% and 2.05% lower accuracy on GSM8K and MATH tasks) due to the high variance of QA pairs extracted from the documents in each cluster. Thus, the QA pairs extracted from the sampled documents cannot well represent the cluster. Similarly, when the cluster number is too high (*i.e.*, $k = 50,000$), there will be many clusters that are in fact contain similar QA pairs, and thus it is hard to explore diverse clusters, thereby leading to the performance degradation (2.56% and 1.79% lower accuracy).

**Warmup Ratio.** As shown in Table 4, we analyzed the impact of the warm-up ratio in `EQUAL`, varying it from 0.1%, 1% to 5% of the candidate document pool $D_c$. Our experiments confirm that using a 1% warm-up ratio of the corpus achieves comparable performance with 5% under lower computation cost.

Table 4: Impact of warm-up ratio on model accuracy.

| Warm-up Ratio | 0.1% | 0.5% | 1% | 3% | 5% |
|---|---|---|---|---|---|
| Accuracy | 69.66% | 71.87% | 72.91% | 72.96% | 73.03% |

**Clustering Algorithms.** Moreover, we evaluate the performance of `EQUAL` by other typical clustering algorithms including BIRCH (Zhang et al., 1996) and DBSCAN (Ester et al., 1996). The details of selecting optimal clustering parameters can be found in the Appendix D. As illustrated in Figure 3(b), `EQUAL` is robust to clustering algorithms on downstream tasks.

## 4.4 Further Experiments

**Chain-of-thought (CoT) Data Generation.** In this section, we consider a setting where, for each QA pair, a corresponding chain-of-thought (CoT) rationale is distilled from a more capable teacher model. In this CoT distillation scenario, we apply our proposed

Table 5: Performance comparison with advanced baselines.

| Method | AIME24 | AIME25 | LiveCodeBench | CodeElo | FLOPs |
|---|---|---|---|---|---|
| Random | 33.1% | 26.0% | 23.2% | 7.29% | 50.65 |
| s1k | 40.7% | 32.7% | 26.7% | 9.30% | 197.87 |
| OpenThoughts | 40.1% | 33.1% | 26.1% | 9.55% | 238.20 |
| EQUAL | **42.7%** | **34.2%** | **27.6%** | **10.05%** | 65.17 |

method `EQUAL` and directly compare it against the state-of-the-art synthetic data generation methods `OpenThoughts` and `s1k` under the same conditions (details of the experimental settings and baseline implementations are provided in Appendix T.). As shown in Table 5, `EQUAL` achieves the best performance while incurring the lowest computational cost (excluding the heuristic `Random`). This advantage stems from performing document-level selection before QA pair extraction and CoT generation, demonstrating the generalizability of `EQUAL`.

**Experiments on Qwen Model.** Moreover, following common practice in recent studies, we conduct experiments on math tasks with Qwen2.5-Math-7B-Instruct (Yang et al., 2024a) and on code tasks with Qwen2.5-Code-7B-Instruct (Hui et al., 2024) (see Appendix T for full experimental details). For evaluation, we include the AIME24 (Zhang & Math-AI, 2024) and

Table 6: Comparison under advanced setting.

| Method | AIME24 | AIME25 | LiveCodeBench | CodeElo |
|---|---|---|---|---|
| Random | 33.1% | 26.0% | 23.2% | 7.29% |
| Avg-sim | 37.1% | 30.7% | 24.7% | 8.04% |
| Perplexity | 37.4% | 31.9% | 24.9% | 8.29% |
| Influence | 35.3% | 29.6% | 23.2% | 7.04% |
| LLM-scoring | 37.6% | 31.3% | 25.1% | 8.54% |
| Rewriting | 39.3% | 30.1% | 24.2% | 7.79% |
| EQUAL | **42.7%** | **34.2%** | **27.6%** | **10.05%** |

AIME25 (Zhang & Math-AI, 2025) benchmarks to assess mathematical reasoning, together with LiveCodeBench (Jain et al., 2024) and CodeElo (Quan et al., 2025) to measure code generation performance. As shown in Table 6, EQUAL consistently outperforms all baselines in this setting, further highlighting its robustness and strong generalization across model families and target capabilities.

# 5    RELATED WORK

**Data Synthesis using LLMs.** In data synthesis, LLMs are commonly used to generate complex and high-quality data (Zhang & Yang, 2023; Yang et al., 2024b) training data. For example, (Luo et al., 2023; Yu et al., 2023) construct LLMs pipeline to revise existing training data, thus enhancing the quality and complexity. (Sun et al., 2024; Wang et al., 2024a) conduct principle-driven prompting, which inserts some well-crafted principles into prompts to guide the LLMs for synthesis. (Guo et al., 2024b; Li et al., 2024c) iteratively synthesize important instruction tuning data and train the model with the newly synthesized data in each round. However, the data generated by these methods often exhibits low diversity, as few-shot prompts tend to make the newly generated data very similar to the original data (Li et al., 2024a; Ding et al., 2024). To address this, researchers have proposed several techniques to generate diverse data. For instance, (Yu et al., 2024a; Gupta et al., 2023) use various prompts to synthesize diverse data. (Yoo et al., 2021) integrates different existing training data to generate more diverse data. (Divekar & Durrett, 2024) uses retrieval augmentation to enhance data diversity by feeding LLMs different retrieved contents. Closer to our work, (Yue et al., 2024) synthesizes QA pairs from vast web documents to enhance the diversity of synthetic data.

**Data Selection for Instruction Tuning.** High-quality data plays a critical role in instruction tuning (Brown, 2020; Zhou et al., 2024; Xia et al., 2024). Simple approaches such as rule-based methods (Soldaini et al., 2024; Penedo et al., 2023) and deduplication (Abbas et al., 2023) can improve data quality but the improvement is limited due to simple heuristics. More sophisticated methods like LLMs like GPT-4 assess data based on human-defined metrics but this is rather expensive (Wettig et al., 2024). Perplexity-based methods (Marion et al., 2023; Li et al., 2024b) select data samples that are difficult for the model to predict. But all above methods do not consider the target distribution of downstream applications. The influence function can measure the impact of training data on downstream model performance (Grosse et al., 2023; Xia et al., 2024), but is computationally intensive and affected by the length of the sequence (Xia et al., 2024), leading to imprecise filtering. (Shao et al., 2024; Yu et al., 2024b) develop a surrogate model trained on available high-quality data to efficiently select data from the candidate dataset; however, its effectiveness is limited by the generalizability of the model.

# 6    CONCLUSION

This paper present EQUAL, a scalable and effective method for instruction tuning data extraction. EQUAL first use contrastive learning to unify the embedding feature spaces of the original documents and the extracted QA pairs. Based on this, EQUAL clusters all candidate document and regards each cluster as an arm of MAB framework due to uncertain distribution similarity scores, allowing sampling from quality clusters to estimate distribution similarity scores accurately while maintaining diversity.

## ACKNOWLEDGMENTS

Chengliang Chai is supported by the NSFC (U25B2019, 62472031), the National Key Research and Development Program of China (2024YFC3308200), Beijing Nova Program, and Huawei. Yuping Wang is supported by the NSFC (U23A20297, U23A20317). Lei Cao is supported by the NSF (DBI-2327954) and Amazon Research Awards

## ETHICS STATEMENT

This work adheres to the ICLR Code of Ethics. In this study, no human subjects or animal experimentation was involved. All datasets used, were sourced in compliance with relevant usage guidelines, ensuring no violation of privacy. We have taken care to avoid any biases or discriminatory outcomes in our research process. No personally identifiable information was used, and no experiments were conducted that could raise privacy or security concerns. We are committed to maintaining transparency and integrity throughout the research process.

## REPRODUCIBILITY STATEMENT

We have made every effort to ensure that the results reported in this paper are fully reproducible. Our experiments are conducted with clearly specified datasets, model architectures, and hyperparameters. All data preprocessing steps, training procedures, and evaluation metrics are explicitly described in the main text and supplementary materials. The codebase used for all experiments will be released publicly upon publication, including scripts for training, evaluation, and data preparation. Additionally, we provide pre-trained models and detailed instructions for reproducing all reported results. We also include sufficient ablation studies and analysis to allow independent verification of our claims.

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

## A    THE USE OF LARGE LANGUAGE MODELS (LLMS)

In the preparation of this work, we used LLM as auxiliary tools in a limited capacity. Specifically, LLMs assisted in drafting portions of the code and in refining the wording of certain sentences for clarity and readability. All technical content, including the design of algorithms, experimental methodology, analysis, and interpretations, was independently developed by the authors. The use of LLMs was confined to language refinement and coding suggestions, and did not influence the scientific contributions or results reported in this paper.

## B    LIMITATIONS

Although `EQUAL` incorporates filtering mechanisms to mitigate some biases, the underlying web corpus may still contain culturally skewed content, which can subtly or unpredictably influence the model's behavior. Specifically, given the vast scale of web-crawled data, it is inevitable that some documents may contain content that is inconsistent with human values (e.g., promoting anti-human or extremist ideologies). Since the focus of our research lies in data selection for targeted capabilities improvement, rather than to detect or filter data with inappropriate or biased content, we did not include strict mechanisms to identify or remove such content, as this falls beyond the scope of our study. Nevertheless, we fully acknowledge the importance of ethical data usage. We therefore recommend that practitioners applying our method first ensure that their candidate data pool has been screened for harmful or culturally biased content (e.g., using bias-detection models such as bert-hateXplain), in order to prevent potential misuse of our approach.

## C    BROADER IMPACT

`EQUAL` mitigates the bias introduced by over-reliance on synthetic or domain-specific data by sourcing instruction tuning data from a broad and diverse web corpus. Its scalable and cost-efficient extraction framework democratizes access to high-quality instruction tuning, enabling researchers with limited resources to fine-tune large language models effectively. Moreover, the enhanced generalization of models trained with `EQUAL` contributes to improved performance in socially impactful domains such as education, healthcare, and public services.

## D    ABLATION STUDY OF CLUSTERING NUMBERS AND ALGORITHM

[*Metric and Criteria.*] We use the metric Within-Cluster Sum of Squares (WCSS) to select the best cluster number using the well-known Elbow (Syakur et al., 2018) algorithm. WCSS is the sum of squared distances between each data instance and its cluster center, i.e., WCSS=$\sum_{i=1}^{k} \sum_{x \in C_i} \|x - \mu_i\|$. At a high level, the criteria should be that within each cluster, data instances are close to each other, based on which it is better for different cluster centers to be far away from each other. Based on the criteria, the Elbow algorithm leverages the WCSS as a measurement to iteratively select an appropriate cluster number, as follows.

[*Specific hypermarameter selection strategy.*] To be specific, Elbow begins with a small $k$, and with $k$ increasing, WCSS first decreases rapidly and then slows down. Then, we identify the "elbow point" where the decreasing rate becomes slow as the best $k$. Thus, within each cluster, data points are sufficiently close to one another. Furthermore, given that $k$ remains modest, different cluster centers tend to maintain a distance from each other.

*Clustering algorithms.* In terms of the clustering algorithms, we also added experiments to show that `EQUAL` is not sensitive to clustering algorithms mainly because different algorithms have their own strategies to select appropriate parameters, which follows the criteria mentioned above.

Specifically, we evaluate the performance of several typical clustering methods including BIRCH (Zhang et al., 1996) and DBSCAN (Ester et al., 1996). Considering that the clustering results are easily affected by the parameters of clustering algorithms, we use different methods to select proper parameters. For DBSCAN, there are 2 key parameters: (1) $eps$(the radius of a neighborhood w.r.t. some data points) and (2) $minPts$ (a data point is considered as a core point if at least $minPts$ data points are within $eps$ of it). They can be set using the method in (Schubert et al.,

2017). For BIRCH (Zhang et al., 1996), we can use the Elbow (Syakur et al., 2018) algorithm or Sihouette score (Shahapure & Nicholas, 2020) to determine the appropriate number of components.

## E    RESTATING THE IMPORTANCE OF OUR PROBLEM SETTING

In this paper, we focus on instruction tuning, also known as supervised fine-tuning (SFT). This phase differs fundamentally from pre-training, whose objective is to build an unaligned foundation model. In contrast, instruction tuning aims to adapt a base LLM to specific domains or user needs, thereby enabling better performance on specialized downstream tasks (Wang et al., 2024b; 2023a; Zhang et al., 2023).

Consequently, many studies on data selection for SFT (Li et al., 2023b; Xia et al., 2024; Ni et al., 2024) emphasize selecting domain-relevant data from a candidate pool with reference to a downstream task. This is because fine-tuning LLMs without careful data selection—for instance, using QA pairs extracted indiscriminately from all documents in the candidate set—can impede the model's ability to acquire specialized capabilities. Empirical results from prior works (Kim et al., 2023; Muennighoff et al., 2022; Wang et al., 2023b) demonstrate that LLMs fine-tuned on targeted subsets of data outperform those trained on the full SFT dataset, underscoring the importance of relevance-aware data selection.

## F    FLOPS CALCULATION

FLOPs is the number of floating point operations performed by GPUs. Many state-of-the-art methods [1,2,3] use it to measure the consumption of GPU computing resources. In our experiments, FLOPs is collected directly in the data selection process using the Python code:

```python
import torch
import torch.nn as nn
from torch.profiler import profile, ProfilerActivity

model = nn.Linear(1024, 512).cuda()
input_data = torch.randn(128, 1024).cuda()
with                      profile(activities=[ProfilerActivity.CPU,
ProfilerActivity.CUDA],
    with_flops=True) as prof:
        model(input_data)
print(prof.key_averages().table(sort_by="flops", row_limit=10))
```

## G    LENGTH OF SELECTED DATA

Table 7: The average length of extracted QA pairs with different methods (*i.e.*, `Influence`, `Perplexity` and `EQUAL(Ours)`)

| Length | Random | Influence | Perplexity | EQUAL(Ours) |
|--------|--------|-----------|------------|-------------|
| Prompt | 48.99  | 37.86     | 49.58      | 58.38       |
| Output | 470.05 | 222.94    | 1235.90    | 438.69      |
| Total  | 519.04 | 260.80    | 1285.48    | 497.07      |

For synthesized QA pair data, recent research (Wang et al., 2022) has shown that having diverse lengths is more beneficial. Short synthesized SFT data may degrade performance due to lacking sufficient context (Taori et al., 2023), while excessively long data can introduce irrelevant content, diminishing the learning signal (Arora et al., 2016).

## H    DETAILED EXPLANATION FOR OPTIMAL TRANSPORT

Optimal transport (OT) is a widely used metric to measure the distribution similarity of semantic vectors in NLP problems (Wang et al., 2022; Chen et al., 2019), which is particularly effective when

the support regions of different distributions have relatively large deviations. In our scenario, the support regions($i.e.$, possible values of the semantic vector) of extracted QA pairs and the target application often diverge due to varying data sources. Thus, traditional divergences, such as KL divergence, are unsuitable because they tends to approach infinity in this situation (Bhardwaj et al., 2021).

Also, in `EQUAL`, OT computation incurs far less cost than QA pair extraction, which dominates the expense of the entire pipeline. This is mainly because extracting QA pairs requires running multiple LLM inferences (e.g., with Qwen2.5-72B) for each document.

| Step | FLOPs |
|---|---|
| Clustering | 2.31 |
| Extracting 5% QA pairs for warm-up | 6.94 |
| Fine-tuning the contrastive learning model | 0.41 |
| **OT calculation** | 0.06 |
| Extracting 5% QA pairs | 6.94 |
| Fine-tuning the LLM on 5% QA pairs | 1.89 |
| **Total** | **18.55** |

Besides, although computing the OT incurs some overhead, it could be negligible compared to the substantial cost savings from avoiding millions of LLM calls. Our experiments (Table 1, Section 4.2) show that `EQUAL` outperforms baselines while reducing QA extraction documents by 5 to 10 times.

## I  ENLARGE THE MODEL SIZE

In this section, we enlarged the model sizes from 7B/8B to 20B. As shown in the Table 8, we observe that `EQUAL` still performs better than other baselines on accuracy because we select high-quality data considering the distribution similarity as a whole. Moreover, `EQUAL` has good scalability compared with other baselines because we use the MAB framework to quickly identify the data instances that are beneficial to the downstream tasks.

Table 8: Enlarge model size and more downstream tasks

| Method | GSM8K | MATH | FLOPs | HUMANEVAL | MBPP | FLOPs | MMLU | BBH | FLOPs |
|---|---|---|---|---|---|---|---|---|---|
| Base | 76.16 | 25.56 | – | 49.6 | 63.0 | – | 67.1 | 70.3 | – |
| Random | 77.33 | 27.61 | 11.23 | 59.7 | 63.8 | 9.67 | 68.7 | 71.3 | 17.65 |
| Avg-sim | 77.97 | 28.33 | 121.25 | 61.4 | 63.0 | 67.73 | 67.6 | 70.4 | 231.02 |
| Perplexity | 76.65 | 31.16 | 156.70 | 62.5 | 63.8 | 73.25 | 68.3 | 72.0 | 302.51 |
| Influence | 76.38 | 29.67 | 511.96 | 62.2 | 64.6 | 373.51 | 70.3 | 73.8 | 976.96 |
| Rewriting | 77.51 | 30.55 | 21.38 | 63.3 | 64.7 | 16.96 | 69.7 | 73.3 | 28.10 |
| Perp.-mab | 77.76 | 31.90 | 21.35 | 63.2 | 63.7 | 17.01 | 69.1 | 72.2 | 28.23 |
| Infl.-mab | 76.70 | 30.73 | 32.76 | 64.7 | 64.1 | 28.46 | 70.6 | 73.7 | 46.53 |
| EQUAL | 80.38 | 33.78 | 21.03 | 67.3 | 66.7 | 16.65 | 73.1 | 76.3 | 27.46 |

## J  STATISTICAL SIGNIFICANCE

In this section, we show the statistical significance of our `EQUAL` in Table 9.

## K  IMPACT AND OPTIMIZATION STRATEGIES OF EMBEDDING MODELS ON QA ALIGNMENT QUALITY

In `EQUAL`, the quality of alignment between documents and QA pairs could be influenced by the embedding model used. Currently, there have been some pretrained embedding models (e.g., jinaai/jina-embeddings-v3, BAAI/bge-large-en-v1.5, OpenAI embeddings and we use BAAI/bge-large-en-v1.5) that demonstrate strong performance across multiple domains. Recent studies show that even in specialized fields such as mathematics and programming, the model's ability to capture

Table 9: Statistical Significance.

| Methods | Exp-1 | Exp-2 | Exp-3 | Avg ± std |
|---|---|---|---|---|
| Random | 68.76 | 68.83 | 69.17 | 68.92 (0.22) |
| Avg-sim | 70.08 | 70.24 | 70.73 | 70.35 (0.34) |
| Perplexity | 63.98 | 64.97 | 64.61 | 64.52 (0.50) |
| Influence | 65.33 | 65.25 | 65.02 | 65.20 (0.16) |
| Rewriting | 64.38 | 64.56 | 64.49 | 64.47 (0.09) |
| Perplexity-MAB | 65.19 | 65.35 | 65.30 | 65.28 (0.08) |
| Influence-MAB | 68.49 | 67.68 | 67.17 | 67.78 (0.67) |
| EQUAL (ours) | 72.51 | 72.70 | 73.78 | 73.01 (0.69) |

general semantics is effective enough to obtain good performance. Furthermore, in `EQUAL`, to improve the embedding model in specific domains, we fine-tune the embeddings on domain-specific QA pairs through the contrastive learning component. This lightweight fine-tuning can significantly enhance representation quality without introducing substantial computational overhead. Although embedding computation does incur some cost, this step is performed offline in our pipeline, allowing the computational burden to be amortized across the entire dataset. In our experiments, this cost is well justified by the downstream performance gains achieved through improved clustering. Overall, while the embedding model plays a critical role, we find its practical impact on clustering quality to be manageable, and the resulting improvements in QA alignment more than justify the associated computational expense.

## L    SAMPLING STRATEGY

In this section, we analyze the impact of different document sampling strategies on downstream performance. In the main text, we primarily adopted Random Sampling as the default strategy for extracting documents from the selected cluster. To thoroughly evaluate the robustness of `EQUAL`, we have conducted additional experiments to evaluate the following three more sophisticated sampling strategies:

Stratified Sampling (Neyman, 1992): Within this strategy, the selected cluster is further partitioned into multiple groups based on their embeddings. Then, a fixed number of samples are drawn from each group, which enables the model to learn from a more comprehensive distribution.

Density Sampling (Palmer & Faloutsos, 2000): We prioritize documents located in high-density regions of the cluster, which is achieved by calculating the local density of each document (e.g., the inverse of $k$-nearest neighbor distance), with the goal of obtaining the most "typical" documents within the cluster.

Diversity Sampling (Carbonell & Goldstein, 1998): Within the selected cluster, we use Maximum Marginal Relevance (MMR) to prioritize documents with the greatest dissimilarity to each other. This strategy aims to maximize the internal diversity of a single cluster.

| Methods | GSM8K Accuracy | HUMANEVAL Accuracy |
|---|---|---|
| Random | 68.7% | 75.2% |
| Stratified | 68.9% | 75.5% |
| Density | 69.4% | 75.7% |
| Diversity | 69.1% | 75.3% |

The experimental results show that `Stratified` Sampling, `Density` Sampling and `Diversity` Sampling achieve slightly higher downstream performance compared to Random Sampling. This suggests that selecting more representative or central documents within a high-quality cluster, as identified by the MAB framework, can marginally improve the efficacy of the extracted data.

However, the performance gain is relatively modest. We attribute this to the inherent nature of our clustering process. After our contrastive learning warm-up step, documents within the same cluster are already highly similar in terms of their potential to generate QA pairs aligned with the target

distribution. The clustering step effectively groups documents with homogeneous characteristics, meaning that the variance of QA pair quality within a single cluster is relatively low. This finding reinforces the importance of our two-stage approach: first, using clustering to create homogeneous groups, and second, using MAB to perform a high-level, cluster-wise exploration-exploitation trade-off.

## M OPEN-ENDED TASKS

To demonstrate the generalizability of `EQUAL`, we conducted an additional experiment on multiple open-ended tasks.

Specifically, we first sampled 500,000 web documents from the KnowledgePile dataset (a well used dataset for general knowledge), forming a heterogeneous and broad candidate data pool $D_c$. Then, we conduct our experiments on Mistral-7B. To better simulate a multi-task instruction tuning scenario, we construct a mixed reference set $D_r$ for `EQUAL`'s selection, combining several publicly available instruction tuning datasets:

- **Summarization**: Summarization task instructions selected from AlpacaFarm (Dubois et al., 2023).
- **Rewriting**: Text rewriting and style transfer task instructions selected from Unnatural Instructions (Honovich et al., 2022).
- **Brainstorming**: Brainstorming and ideation task instructions selected from the "brainstorming" and "ideation" categories of FlanV2 (Longpre et al., 2023).

For evaluation, we use `AlpacaEval` (Dubois et al., 2024), a GPT-4-based benchmark for assessing a model's performance on open-ended instruction following tasks (e.g., summarization, rewriting, creative writing) and the primary evaluation metric is the model's win rate (Dubois et al., 2024) against the baseline (i.e., the original Mistral-7B base model), as judged by GPT-4 based on response quality.

| Methods/Win rate(%) | Summarization | Rewriting | Brainstorming | Average |
|---|---|---|---|---|
| Random | 70.9 | 70.9 | 54.7 | 65.5 |
| Avg-sim | 73.5 | 71.2 | 56.3 | 67.0 |
| Perplexity | 75.1 | 73.3 | 57.6 | 68.7 |
| Influence | 74.0 | 73.5 | 57.4 | 68.3 |
| LLM-scoring | 76.2 | 74.0 | 58.3 | 69.5 |
| Rewriting | 75.5 | 74.2 | 57.9 | 69.2 |
| Perplexity-MAB | 73.5 | 73.1 | 56.5 | 67.7 |
| Influence-MAB | 72.0 | 71.8 | 55.7 | 66.5 |
| EQUAL | 78.1 | 77.1 | 60.5 | 71.9 |

The results clearly indicate that the `EQUAL` method is particularly effective for open-ended tasks and substantially improves the target model's capability to manage multiple tasks concurrently. Specifically, the model fine-tuned with `EQUAL` achieved an average win rate of 71.9% on `AlpacaEval`, significantly outperforming the random selection baseline (+6.4%) and the Perplexity-MAB baseline (+4.2%). This provides strong evidence that `EQUAL` can effectively identify the most valuable content from a large document pool to simultaneously enhance multiple open-ended task capabilities.

Specifically, in these experiments, we focus on open-ended tasks—such as summarization, rewriting, and brainstorming, which lack clearly defined ground-truth answers. To address the challenge of evaluating such tasks, we follow the approach proposed in AlpacaEval, which leverages powerful large language models (LLMs) to assess generated responses without relying on reference outputs. Specifically, both our fine-tuned model and a baseline model (i.e., the original base model used in our experiments) are prompted with the same open-ended questions. A high-performing LLM (e.g., GPT-4) is then used to compare their responses and determine which one is better. Following AlpacaEval, we report the win rate of our model over the baseline across all open-ended questions as our primary accuracy metric. Importantly, the default evaluation dataset in AlpacaEval does not cover the tasks of summarization, rewriting, or brainstorming. Therefore, we used (See et al., 2017;

Wieting & Gimpel, 2017; Lin et al., 2019) as the evaluation datasets for these three tasks, respectively, in our experiments.

## N  DIVERSITY ANALYSIS

In this section, we directly assess diversity to further validate the effectiveness of EQUAL. Specifically, we measure the lexical diversity and semantic diversity among the extracted QA pairs.

1. Lexical Diversity: We compute Type-Token Ratio (TTR) and Measure of Textual Lexical Diversity (MTLD) over the QA pairs extracted by each method. As shown in the table below, EQUAL achieves the highest TTR and MTLD scores, indicating richer vocabulary usage in the extracted QA pairs.

| Methods | TTR | MTLD |
|---|---|---|
| Random | 55% | 62.0% |
| Perplexity | 46% | 53.3% |
| Influence | 42% | 49.1% |
| EQUAL | 52% | 61.7% |

2. Semantic Diversity: We compute the average pairwise semantic similarity among embeddings (using BAAI/bge-en-v1.5) within the final extracted dataset. Lower average similarity indicates higher semantic diversity. The results in the table below indicate that the QA pairs extracted by EQUAL exhibit greater semantic diversity.

| Methods | Cosine Similarity |
|---|---|
| Random | 50% |
| Perplexity | 56% |
| Influence | 59% |
| EQUAL | 51% |

These results demonstrate that EQUAL constructs a more lexically and semantically diverse instruction dataset.

## O  COMPARE WITH OTHER DATASETS

To directly evaluate the quality of data constructed by EQUAL, we compare it against four leading math-focused instruction-tuning datasets — MathInstruct[1], MetaMathQA[2], XwinMath[3], and OpenMathInstruct[4] — using full fine-tuning on LLaMA-3-8B under identical settings. As shown in the table below, EQUAL achieves the highest accuracy on both the GSM8K and MATH benchmarks, while maintaining comparable FLOPs to other methods, demonstrating its effectiveness. Specifically, all the four baselines use LLM to rewrite and augment the QA seed data, but limited data diversity leads to their inferior performance compared to EQUAL.

[1] Mammoth: Building math generalist models through hybrid instruction tuning

[2] Metamath: Bootstrap your own mathematical questions for large language models

[3] Common 7b language models already possess strong math capabilities

[4] Openmathinstruct-1: A 1.8 million math instruction tuning dataset

| Methods | GSM8K | MATH | FLOPs |
|---|---|---|---|
| MathInstruct | 67.30% | 31.33% | 18.25 |
| MetaMathQA | 69.23% | 33.02% | 17.76 |
| XwinMath | 69.72% | 33.79% | 17.97 |
| OpenMathInstruct | 70.03% | 33.53% | 18.31 |
| EQUAL | 73.01% | 35.10% | 18.55 |

## P    REFERENCE DATA SIZE

Intuitively, more reference data leads to more accurate estimation of the target distribution. However, in this section, we would like to clarify that as long as the reference data roughly capture the distribution, EQUAL can perform effectively in practice. This is consistent with the results reported in other papers on this issue — (Xia et al., 2024; Li et al., 2023b; Wang et al., 2023b) all perform well using only dozens of validation examples. To further validate this, we conducted an additional experiment on the math task using full fine-tuning of LLaMA-3-8B by varying the size of the reference set. The results are summarized in the table below:

| #-Reference data | 20 | 50 | 100 | 500 | 1000 | 1500 |
|---|---|---|---|---|---|---|
| Accuracy | 73.01% | 73.08% | 73.15% | 73.21% | 73.27% | 73.33% |

As shown in the table, the performance of EQUAL improves consistently with the increase in the number of examples in the reference set. Note that EQUAL achieves good results with just 20 reference examples, reaching an accuracy of 73.01%—nearly matching the 73.33% obtained with 1500 examples. Furthermore, we also extended our analysis to more complex and diverse benchmarks such as MATH. The results show that, for more challenging dataset, the OT-based distance estimation indeed requires 150 reference samples to reach stable and effective performance. In practice, such reference sets are often readily available (e.g., training set in benchmark tasks), making our method broadly applicable.

| #-Reference data | 20 | 50 | 75 | 150 | 500 | 1000 | 1500 |
|---|---|---|---|---|---|---|---|
| Accuracy | 34.37% | 35.10% | 35.76% | 36.11% | 36.19% | 36.23% | 36.40% |

## Q    PERFORMANCE ON MORE DIFFICULT TASKS

In this section, we evaluate the effectiveness of EQUAL on the more challenging math benchmarks AIME2024 and OLYMPIABENCHMATH, as well as the code benchmarks HUMANEVAL+ and LCB. The results are presented below:

| Methods | AIME2024 | OLYMPIABENCHMATH | HUMANEVAL+ | LCB |
|---|---|---|---|---|
| Llama-3-8B | 1.1% | 3.7% | 31.1% | 3.6% |
| Llama-3-8B-Instruct | 8.3% | 14.4% | 60.4% | 9.7% |
| EQUAL | 10.1% | 17.3% | 61.8% | 10.4% |

## R    BASELINE IMPLEMENTATION DETAILS

In our paper, the perplexity scores are computed using an off-the-shelf language model (i.e., our target model) that has not been fine-tuned on the reference set. Here, we include perplexity scores computed on a model fine-tuned with the reference set as an informative additional baselines. Specifically, we introduced an improved baseline called perplexity-ref. In this approach, we first finetune the perplexity model on the reference set $D_r$, enabling it to better capture the specific domain characteristics of the downstream task. We then use this finetuned model to compute perplexity scores for all QA pairs extracted from the candidate pool to further finetune the model.

Experimental results show that perplexity-ref outperforms the standard perplexity method. This is because the finetuned model is more aligned with the downstream task, making the perplexity scores more indicative of the relevance of the data. However, perplexity-ref does still not perform better than our proposed EQUAL, as it treats data points independently and fails to capture the underlying relationships between them, which in turn leads to reduced data diversity.

Besides, in our experiments, we used the same method as LESS (Xia et al., 2024) to calculate the influence scores on our target model. The reference set and test set are consistent with that used in our EQUAL method. The training set consists of QA pairs extracted from all documents, from which we select the top 5% with the highest influence function scores calculated against the reference set.

| Methods | GSM8K | MATH | HUMANEVAL | MBPP |
|---|---|---|---|---|
| Perplexity | 64.52% | 33.56% | 44.5% | 50.5% |
| Perplexity-ref | 69.10% | 34.46% | 47.6% | 54.6% |
| EQUAL | 73.01% | 35.10% | 49.4% | 56.3% |

## S  EXPERIMENTS ON QWEN MODEL

We conducted additional experiments using the Qwen2.5-7B model. The results in the table below show that our proposed method consistently improves performance on Qwen2.5-7B, demonstrating its strong generalization across different model architectures.

| Methods | GSM8K | MATH | FLOPs | HUMANEVAL | MBPP | FLOPs |
|---|---|---|---|---|---|---|
| Random | 86.1% | 59.6% | 8.51 | 66.7% | 75.2% | 6.73 |
| Avg-sim | 87.6% | 65.2% | 111.73 | 68.7% | 75.9% | 64.15 |
| Perplexity | 86.1% | 66.7% | 146.52 | 69.1% | 76.1% | 72.22 |
| Influence | 85.6% | 61.0% | 251.28 | 66.9% | 75.3% | 131.65 |
| LLM-scoring | 86.9% | 67.1% | 17.61 | 69.6% | 76.4% | 137.96 |
| Rewriting | 87.1% | 61.3% | 19.06 | 67.2% | 75.2% | 14.56 |
| Perplexity-MAB | 86.4% | 67.0% | 18.23 | 69.7% | 76.6% | 14.19 |
| Influence-MAB | 85.9% | 61.6% | 18.19 | 67.8% | 75.5% | 14.72 |
| EQUAL | 89.6% | 71.3% | 17.76 | 73.3% | 78.0% | 13.30 |

## T  CoT DATA GERERATION.

In this section, we further consider an experimental setting where, for each document selected from the candidate pool, we extract its QA pair and distill a corresponding chain-of-thought (CoT) for that pair using a stronger teacher model. In this CoT distillation scenario, we apply our EQUAL and directly compare it against OpenThoughts and s1k as follows:

1) OpenThoughts: We first extract QA pairs from all candidate documents and employ QwQ-32B to generate chain-of-thought answers for each question, and then use the resulting data to train our target model.

2) s1K: We first extract QA pairs from all candidate documents and use QwQ-32B to generate chain-of-thought answers for them. Subsequently, we follow the s1K methodology to select a small subset of QA pairs by filtering for quality, difficulty, and diversity.

3) EQUAL: We employ a multi-armed bandit strategy to iteratively identify clusters that are most likely to yield high-quality CoT data points. We then restrict expensive QwQ-32B calls to the documents in these high-value clusters only to extract QA pairs and generate chain-of-thought answers for them.

| Methods | AIME24 | AIME25 | LiveCodeBench | CodeElo | FLOPs |
|---|---|---|---|---|---|
| Random | 33.1% | 26.0% | 23.2% | 7.29% | 50.65 |
| s1k | 40.7% | 32.7% | 26.7% | 9.30% | 197.87 |
| OpenThoughts | 40.1% | 33.1% | 26.1% | 9.55% | 238.20 |
| EQUAL | 42.7% | 34.2% | 27.6% | 10.05% | 65.17 |

As shown in the table above, aside from the heuristic method Random, EQUAL attains the best performance with the lowest computational cost (i.e., FLOPs), primarily because it performs document-level selection before QA extraction and CoT generation. Specifically, instead of extracting QA pairs from all documents, EQUAL treats document clusters as arms in a multi-armed bandit and iteratively selects the most promising ones for QA extraction and CoT generation. This exploration–exploitation scheme focuses extraction on clusters that yield larger gains on targeted capabilities, cutting LLM extraction cost while preserving or even improving final task performance.

## U    EXTRACTOR MODEL.

In this part, we evaluate the performance of EQUAL using extractor models of varying sizes, as well as with closed-source extractor models. Specifically, instead of the Qwen2.5-72B model used in our paper, we performed extraction with the weaker-extraction Qwen2.5-7B and the closed-source Qwen-Flash models. As shown in Figure 4, replacing the Qwen2.5-72B extractor with the Qwen2.5-7B model or the closed-

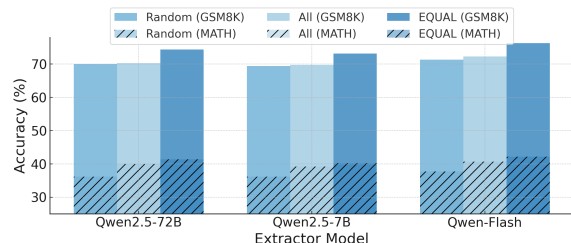

Figure 4: Ablation study of different extractor model.

source Qwen-Flash model leads to small shifts in absolute performance (a decrease for Qwen2.5-7B and an increase for Qwen-Flash), attributable to differing linguistic and reasoning abilities, the relative improvement trend introduced by EQUAL remains consistent. This indicates that the key strength of our method lies in its ability to mine high-quality QA pairs from large-scale web data, rather than in the ability of the extractor itself.

## V    KEY DIFFERENCE BETWEEN EQUAL AND EXISTING METHODS

In Existing methods, when an LLM is prompted with only a few seed examples, it tends to produce instructions that are locally similar to those seeds (i.e., few-shot examples), which limits diversity. In contrast, all QA pairs generated by EQUAL are extracted from a large, heterogeneous web corpus (AutoMathText, KnowledgePile, StackOverflow) that contains rich knowledge beyond the seeds. Thus, whereas existing methods literally "stay close" to the seed examples in input space, EQUAL selects real, naturally occurring QA pairs from an independent corpus and uses the seeds solely as a guidance signal.

Furthermore, EQUAL prevents the generated QA pairs from collapsing onto a few high-OT clusters by modeling document selection as a multi-armed bandit (MAB) strategy with an upper-confidence-bound (UCB) style exploration term. Clusters that have been sampled less often receive a larger bonus and are thus more likely to be selected. This explicitly promotes under-sampled clusters, encouraging us to explore new regions of the document space instead of repeatedly pulling only the highest-OT cluster, which further increases the diversity of the generated QA pairs.

Together, our method avoids generating QA pairs that are near-duplicates of the original seed examples by merely rewriting a small set of seed examples. Instead, we selecting naturally generated QA pairs from large, heterogeneous corpora whose distribution matches the target tasks.

## W    PROMPTS FOR QA PAIR EXTRACTION

In this section, we present the prompts we used for extraction tasks across different domains.

### W.1    CODING TASK

```
Code

SYSTEM:

You are given a set of pre-processed documents, each of which may contain natural question-answer
(Q-A) pairs. Your task is to identify and extract these pairs while ignoring unrelated content such as
ads, markup, or boilerplate text.
Input:
Each document contains multiple sections of text. Some of these sections may have clear questions
followed by answers, while others may be irrelevant (e.g., ads or noise).
Output:
```

Extract the Q-A pairs found within each document. A valid Q-A pair must consist of a clearly defined question and its corresponding answer. If no natural Q-A pair exists in the document, return void for that document. In the document, in order to describe the problem more clearly, the questioner usually attaches some useful information (e.g., code or explaination) to make it easier for others to better understand the problem. You need to extract this part of the content that needs to be complete as well.

Here are some examples:

# Example 1

Content:

Sorting lines date-wise and time-wise using Python from a `.txt` file. I have just written a Python code to extract data from around 700 text files into one file called `out_data.txt`. The contents of the `out_data.txt` file look something like this:

```
datetime,V_1,V_2,V_3,V_4,V_5,V_6,V_7
2013-03-17 18:01:48.372,100,884,776,009,6553,ffff,987
2013-03-17 18:02:03.828,876,632,887,008,5423,879,443
2013-05-17 20:13:52.488,543,987,233,112,098,344,123
2013-08-17 23:09:08.171,667,9887,9897,09876,0987,098,0987
2013-01-17 35:06:04.172,267,987,6897,9876,1287,3498,2987
...
```

There are a total of 5,783,374 lines in the `out_data.txt` file, and each line (after the header) begins with the datetime value.

However, the problem I have is that the code I wrote extracts the data from each individual file and adds it to my `out_data.txt` file, but the lines are not in the order of date-time as you can see above. I was hoping to get my lines to be in date-time order because I need to plot this data. Any help will be highly appreciated !

Here is my current code:

```python
import re   # Regular expressions
import glob   # File management and reading

if __name__ == "__main__":   # Opening for Python
    all_header = []   # List declaration
    all_values = []   # List declaration
    i = 0
    with open('out_data.txt', 'w') as of:   # Output file
        for infile in glob.glob("/Users/name/Desktop/raw_data/*.txt"):   # ↪ Input file
            with open(infile) as fobj:
                print(f"Processing file {infile}")
                for line in fobj:
                    data = line.split()   # Split each line into ↪ individual tokens
                    if len(data) == 2 and re.search(r'(\d+-\d+-\d+)', ↪ data[0]):   # Regular expression to identify date ↪ and time
                        header = ['datetime']   # Column name datetime
                        values = [data[0] + " " + data[1]]   # date+time ↪ as one value
                    else:
                        header = [d for d in data if data.index(d) % 2 == ↪ 0]
                        values = [d for d in data if data.index(d) % 2 != ↪ 0]
                    all_header.extend(header)
                    all_values.extend(values)
                    if not header:
                        if i == 0:
                            of.write(','.join(all_header))
                        i += 1
                        of.write("\n")
                        of.write(','.join(all_values))
```

```
                          all_header = []
                          all_values = []
        of.write("\n")
        of.write(','.join(all_values))
```

EXPECTED RESULT

The expected result from the example data given above would be:

```
datetime ,V_1,V_2,V_3,V_4,V_5,V_6,V_7
2013−01−17  35:06:04.172,267,987,6897,9876,1287,3498,2987
2013−03−17  18:01:48.372,100,884,776,009,6553,ffff,987
2013−03−17  18:02:03.828,876,632,887,008,5423,879,443
2013−05−17  20:13:52.488,543,987,233,112,098,344,123
2013−08−17  23:09:08.171,667,9887,9897,09876,0987,098,0987
```

However, I could not figure out how to include the sort element in the code or if there is any other way to achieve this.

SOLUTION USING PANDAS

You can use `pandas`. A simple example would be as follows:

```python
import pandas as pd
import glob

df_list = []
for infile in glob.glob("/Users/name/Desktop/raw_data/*.txt"):
    df_list.append(pd.read_csv(infile, parse_dates=['datetime']))
df = pd.concat(df_list).sort_values(by='datetime')
df.to_csv('out_data.txt', index=False)
```

SOLUTION USING CSV

An alternative method is:

```python
import csv

with open("out_data.txt", "r") as f:
    reader = csv.reader(f, delimiter=",")
    header = next(reader)
    sortedlist = sorted(reader, key=lambda x: x[0])

with open("sorted.txt", "w") as f:
    writer = csv.writer(f, lineterminator="\n")
    writer.writerow(header)
    writer.writerows(sortedlist)
```

SOLUTION USING BASH

As an alternative, you can also use Bash:

```bash
head −1 out_data.txt > sorted.txt
tail +2 out_data.txt | sort −t, −k1 >> sorted.txt
```

Hope this helps.

## Q:

I've just written a Python code to extract data from around 700 text files into one file called `out_data.txt`.
The contents of the `out_data.txt` file look something like this:

```
datetime ,V_1,V_2,V_3,V_4,V_5,V_6,V_7
2013−03−17  18:01:48.372,100,884,776,009,6553,ffff,987
2013−03−17  18:02:03.828,876,632,887,008,5423,879,443
2013−05−17  20:13:52.488,543,987,233,112,098,344,123
```

```
2013−08−17 23:09:08.171,667,9887,9897,09876,0987,098,0987
2013−01−17 35:06:04.172,267,987,6897,9876,1287,3498,2987
...
```

There are a total of 5,783,374 lines in the `out_data.txt` file, and each line (after the header) begins with the `datetime` value.

However, the problem I have is that the code I wrote extracts the data from each individual file and adds it to my `out_data.txt` file, but the lines are not in the order of `datetime` as you can see above. I was hoping to get my lines to be in `datetime` order because I need to plot this data. Any help will be highly appreciated.

A:

```python
import re   # regular expressions
import glob   # file management and reading

if __name__ == "__main__":   # opening for Python
    all_header = []   # list declaration
    all_values = []   # list declaration
    i = 0
    with open('out_data.txt', 'w') as of:   # output file
        for infile in glob.glob("/Users/name/Desktop/raw_data/*.txt"):   #
        ↪  input files
            with open(infile) as fobj:
                print("processing file {}".format(infile))
                for line in fobj:
                    data = line.split()   # split each line into
                    ↪  individual tokens
                    if len(data) == 2 and re.search(r'(\d+-\d+-\d+)',
                    ↪  data[0]):   # identify date
                        header = ['datetime']   # column name
                        values = [data[0] + " " + data[1]]   # combine
                        ↪  date and time
                    else:
                        header = [d for d in data if data.index(d) % 2 ==
                        ↪  0]
                        values = [d for d in data if data.index(d) % 2 !=
                        ↪  0]
                    all_header.extend(header)
                    all_values.extend(values)
                    if not header:
                        if i == 0:
                            of.write(','.join(all_header))
                        i += 1
                        of.write("\n")
                        of.write(','.join(all_values))
                        all_header = []
                        all_values = []
        of.write("\n")
        of.write(','.join(all_values))
```

EXPECTED RESULT:

From the example data given above, the output should be:

```
datetime ,V_1,V_2,V_3,V_4  ,V_5  ,V_6   ,V_7
2013−01−17 35:06:04.172,267 ,987 ,6897,9876,1287,3498 ,2987
2013−03−17 18:01:48.372,100 ,884 ,776 ,009 ,6553,ffff ,987
2013−03−17 18:02:03.828,876 ,632 ,887 ,008 ,5423,879  ,443
2013−05−17 20:13:52.488,543 ,987 ,233 ,112 ,098 ,344  ,123
2013−08−17 23:09:08.171,667 ,9887,9897,09876,0987,098  ,0987
```

USING PANDAS:

You can use the Pandas library for simplicity. Here is an example:

```python
import pandas as pd
import glob

df_list = []
for infile in glob.glob("/Users/name/Desktop/raw_data/*.txt"):
    df_list.append(pd.read_csv(infile, parse_dates=['datetime']))
df = pd.concat(df_list).sort_values(by='datetime')
df.to_csv('out_data.txt', index=False)
```

USING CSV MODULE:

You can also perform an ordinary (dictionary order) sort as follows:

```python
import csv

with open("out_data.txt", "r") as f:
    reader = csv.reader(f, delimiter=",")
    header = next(reader)
    sortedlist = sorted(reader, key=lambda x: x[0])

with open("sorted.txt", "w") as f:
    writer = csv.writer(f, lineterminator="\n")
    writer.writerow(header)
    writer.writerows(sortedlist)
```

USING BASH:

Alternatively, you can use the following Bash commands:

```bash
head -1 out_data.txt > sorted.txt
tail +2 out_data.txt | sort -t, -k1 >> sorted.txt
```

Hope this helps!

## W.2 MATHEMATICAL TASK

**Math**

**System:**
You are an excellent AI assistant who is good at constructing question-answer (Q-A) pairs. Your task is to construct some math Q-A from the original documents.
Input:
Each document contains multiple sections of text. Some of these sections may contain mathematical content which can be used to construct Q-A pairs.
Output:
Identify valid content and construct Q-A pairs. A valid Q-A pair must consist of a clearly defined question and its corresponding answer. Specially, the questions should be solvable that provide valid and complete pre-conditions; and the answers need to satisfy the Chain of Thought (CoT) format, which instructs the responder to solve the question step by step. If the content in the document is not suitable for Q-A construction, return void for that document.
Here is an example:
**User**: As I mentioned certain scientific terms in my previous post, I would like to go in-depth on those concepts, beginning with terminal velocity, it being the most fundamental concept in my post.
**So what is terminal velocity?**
Terminal velocity is the velocity of an object when the drag force (dependent on the fluid the object is travelling through) acting upon it is equal to the downward force of gravity acting upon it. Simply put, when the air resistance of a falling object cancels out the gravitational force which is pulling it downwards and accelerating it.

**So how do these forces affect the motion of the object?** The forces cancelling each other out make the object remain at a constant rate of motion.

You may ask why does the object still move when the forces cancel each other out. This is due to the fact that in the beginning the force of gravity still manages to overcome the drag force, allowing the object to gain speed (accelerate) initially. But as the object increases in velocity, the drag force increases. This effect can also be seen in the case of friction (Drag and friction are pretty much the same thing). Let's assume that a boy is dragging a heavy box, full of files, across a distance of 100 meters. Now, we will imagine this scenario in two different ways: firstly, in the case whereby the boy is walking slowly, and in the second, whereby the boy is running. So in the first case, the boy walks; when he reaches the end, he feels the bottom of the box, where the box and the floor meet, it still feels the same as before. Now in the second case, he runs; he once again feels the bottom of the box, this time it feels warmer than before.

**So what can we infer from this scenario?**

Before I reveal the answer, I would like to state a few properties of friction:

- Friction opposes motion

- Friction causes wear and tear

- Friction produces heat when kinetic energy is converted into thermal energy

**So what can we infer?** In the second scenario, there was more heat; therefore, we can assume that there was more frictional force produced in the second case.

Now let's go back to what I mentioned previously, air resistance increases (Drag Force) as the object's velocity increases. As seen in the example above, we can tell that this statement is true.

**Recap**:

- Terminal velocity is the velocity an object is at when the gravitational force acting upon it is equal to the drag force acting upon it in the opposite direction, therefore cancelling out all forces and resulting in a resultant force of 0.

- The drag force acting upon the object increases as the object accelerates due to the downward force of gravity.

Ok, so let's move on to the math behind terminal velocity and some examples of it.

The formula for terminal velocity is as follows:

$$V_t = \sqrt{\frac{2mg}{\rho A C_d}}$$

where:

- $m$ = Mass of falling object

- $g$ = Acceleration of the object due to gravity

- $\rho$ = Density of fluid through which the object is travelling

- $A$ = Projected area of the object

- $C_d$ = Drag Coefficient

Example:

Assuming I drop a metal cube which has a mass of $3\,\text{kg}$ and has a projected area of $1\,\text{m}^2$ on Earth $90°$ downward, through air at a temperature of $25°$C, what would the terminal velocity of the cube be?

All we have to do is input all the values into the formula. The acceleration due to gravity on Earth is $9.81\,\text{m/s}^2$. The density of air at $25°$C is $1.1839\,\text{kg/m}^3$ and the drag coefficient of a cube is $1.05$ facing downward.

The result is:

$$V_t = 6.881101581\,\text{m/s}.$$

That's terminal velocity for you!

I would like to thank Mr. Tan Ping Hock and Mr. Yao Zhi Wei Adrian, my current and previous physics teachers respectively, for clearing my doubts about certain concepts within this topic of terminal velocity!

Thanks for reading!

