# OpenReview forum: "Not All Documents Are What You Need for Extracting Instruction Tuning Data"
_ICLR.cc/2026/Conference — ICLR 2026 Poster_

### Official Review · Reviewer_wiHJ · 2025-10-29

**Soundness:** 3
**Presentation:** 3
**Contribution:** 3
**Rating:** 6
**Confidence:** 4

**Summary:**

The paper focuses on efficient instruction-tuning data curation from unstructured text in the documents. Specifically, it argues that the existing methods generate many QA pairs from all the documents using LLMs which is computationally prohibitive. In addition, many of them suffer from low-quality data problems. To fix this, the paper proposes an iterative solution, posed as a multi-arm bandit problem, where they first cluster all documents in the embedding space and quantify their quality based on the optimal transport distance between the QA pairs extracted from them and reference datasets. Further, they compute the diversity score of each cluster so that the QA extraction is focused on exploration first and exploitation later. They perform several experiments to show the effectiveness of the method on llama-3.1-8b and mistral models on math and code tasks. Further, they perform several ablations to show the usefulness of diverse design choices.

**Strengths:**

1. The limitation pointed out by the work makes sense, and the proposed solution is tailored to fix them. In particular, the algorithm is reasonable which poses document selection and QA pair generation as a multi-bandit problem.

2. The paper also proposes a contrastive learning approach to ensure that the features extracted from the documents and extracted QA pairs are similar in the representation space while they are far apart for original documents and negative QA pairs.

3. The experimental results are quite extensive in terms of baselines and ablation studies. It definitely shows the general usefulness of the approach.

**Weaknesses:**

1. Line 46-48 argues that the synthetic data generation LLMs suffer from lack of diversity because of their proximity to the seed examples. The proposed method may also suffer from the same problem since the algorithm is optimized to reduce the optimal transport distance between the QA pairs from the documents and SEED datasets (MATH/GSM-8K or MBPP).

2. The choice of models and evaluation datasets is somewhat old. For instance, it is quite common to show some quantitative experiments on Qwen models nowadays. In addition, there are more challenging datasets such as AIME24, 25 for math and LiveCodeBench or CodeElo for Code.

3. There is no comparison against state-of-the-art synthetic data generation methods such as OpenThoughts, s1k etc.

**Questions:**

Mentioned above

---

> ### Author Response · Authors · 2025-11-21
>
> $\textbf{Overall Response:}$
>
> Many thanks for your insightful feedback! We have added 2 additional experiments and provided sufficient discussion w.r.t. your concerns about the diversity of generated data, additional experimental settings and comparison with advanced baselines, etc. We have also included these experiments and discussion into the revised paper and promise to include them into the final version if the paper is accepted.
>
> $\textbf{Response to Weakness 1:}$
>
> Nice comment! We apologize that our description in the introduction was not precise. In fact, the key issue of existing methods is that their seed examples lack diversity, whereas our approach avoids this issue by using documents from large-scale web corpora as seed documents.
>
> Specifically, when an LLM is prompted with only a few seed examples, it tends to produce instructions that are locally similar to those seeds (i.e., few-shot examples), which limits diversity. In contrast, all QA pairs generated by `EQUAL` are extracted from a large, heterogeneous web corpus (AutoMathText, KnowledgePile, StackOverflow) that contains rich knowledge beyond the seeds. Thus, whereas existing methods literally ``stay close'' to the seed examples in input space, `EQUAL` selects real, naturally occurring QA pairs from an independent corpus and uses the seeds solely as a guidance signal.
>
> Furthermore, `EQUAL` prevents the generated QA pairs from collapsing onto a few high-OT clusters by modeling document selection as a multi-armed bandit (MAB) strategy with an upper-confidence-bound (UCB) style exploration term. Clusters that have been sampled less often receive a larger bonus and are thus more likely to be selected. This explicitly promotes under-sampled clusters, encouraging us to explore new regions of the document space instead of repeatedly pulling only the highest-OT cluster, which further increases the diversity of the generated QA pairs.
>
> Together, our method avoids generating QA pairs that are near-duplicates of the original seed examples due to merely rewriting a small set of seed examples. Instead, we select naturally generated QA pairs from large, heterogeneous corpora whose distribution matches the target tasks. In the revised manuscript, we have updated the introduction (Lines 45-47) and added the discussion details into appendix U to more clearly demonstrate the difference between `EQUAL` and existing methods.
>
> $\textbf{Response to Weakness 2:}$
>
> We thank the reviewer for raising this important point and have run new experiments with more recent models and challenging benchmarks. Specifically, following common practice in recent studies, we conduct experiments on math tasks with Qwen2.5-Math-7B-Instruct and on code tasks with Qwen2.5-Code-7B-Instruct. In addition, we incorporate experiments on the AIME24/25 benchmark to evaluate mathematical reasoning capabilities, as well as on the LiveCodeBench and CodeElo benchmarks to assess code generation performance. The new results shown in the tables below have been added to Section 4.4 of the revised manuscript, further demonstrating the robustness and generality of our `EQUAL` across different model families and targeted capabilities.
>
> | Methods        | AIME24 | AIME25 | LiveCodeBench | CodeElo |
> |---------------|-------:|-------:|--------------:|--------:|
> | `Random`      | 33.1%  | 26.0%  | 23.2%         | 7.29%   |
> | `Avg-sim`     | 37.1%  | 30.7%  | 24.7%         | 8.04%   |
> | `Perplexity`  | 37.4%  | 31.9%  | 24.9%         | 8.29%   |
> | `Influence`   | 35.3%  | 29.6%  | 23.2%         | 7.04%   |
> | `LLM-scoring` | 37.6%  | 31.3%  | 25.1%         | 8.54%   |
> | `Rewriting`   | 39.3%  | 30.1%  | 24.2%         | 7.79%   |
> | `EQUAL`       | 42.7%  | 34.2%  | 27.6%         | 10.05%  |

---

> ### Author Response · Authors · 2025-11-21
>
> $\textbf{Response to Weakness 3:}$
>
> Thank you for this helpful suggestion! Following the state-of-the-art methods  mentioned by the reviewer, we further consider an experimental setting where for each document selected from the candidate pool, we extract its QA pair and distill a corresponding chain-of-thought (CoT) for that pair using a stronger teacher model. In this CoT distillation scenario, we apply our `EQUAL` and directly compare it with `OpenThoughts` and `s1k` as follows:
>
>
> 1) `OpenThoughts`: we first extract QA pairs from all candidate documents and employ QwQ-32B to generate chain-of-thought answers for each question, and then use the resulting data to train our target model.
>
> 2) `s1K`: we first extract QA pairs from all candidate documents and use QwQ-32B to generate chain-of-thought answers for them. Subsequently, we follow the `s1K` methodology to select a small subset of QA pairs by filtering for quality, difficulty, and diversity.
>
> 3) `EQUAL`: we employ a multi-armed bandit strategy to iteratively identify clusters that are most likely to yield high-quality CoT data points. We then restrict expensive QwQ-32B calls to the documents in these high-value clusters only to extract QA pairs and generate chain-of-thought answers for them.
>
> | Methods          | AIME24 | AIME25 | LiveCodeBench | CodeElo | FLOPs  |
> |-----------------|-------:|-------:|--------------:|--------:|-------:|
> | `Random`        | 33.1%  | 26.0%  | 23.2%         | 7.29%   | 50.65  |
> | `s1k`           | 40.7%  | 32.7%  | 26.7%         | 9.30%   | 197.87 |
> | `OpenThoughts`  | 40.1%  | 33.1%  | 26.1%         | 9.55%   | 238.20 |
> | `EQUAL`         | 42.7%  | 34.2%  | 27.6%         | 10.05%  | 65.17  |
>
>
> As shown in the table above, aside from the heuristic method `Random`, `EQUAL` attains the best performance with the lowest computational cost (i.e., FLOPs), primarily because it performs document-level selection before QA extraction and CoT generation. Specifically, instead of extracting QA pairs from all documents, `EQUAL` treats document clusters as arms in a multi-armed bandit and iteratively selects the most promising ones for QA extraction and CoT generation. This exploration–exploitation scheme prioritizes extraction on clusters that yield larger gains on targeted capabilities, cutting LLM extraction cost while preserving or even improving the final task performance.
>
> We have updated the manuscript accordingly (as detailed in Section 4.4) to include these experiments and the related discussion.

---

> ### Author Response · Authors · 2025-11-26
> **Kindly Request to Reviewer wiHJ**
>
> Dear reviewer wiHJ,
>
> Wishing you a happy and blessed Thanksgiving!
>
> We would like to express our gratitude to your valuable feedback. We have carefully considered all suggestions and updated our submission accordingly.
>
> If you have any question for our paper, please feel free to point out and we will try to address it as soon as possible. We would like to express our sincere gratitude for your valuable comments again. Thanks and looking forward to your reply!
>
> Best wishes!
>
> Authors

---

### Official Review · Reviewer_5USS · 2025-10-30

**Soundness:** 2
**Presentation:** 3
**Contribution:** 2
**Rating:** 4
**Confidence:** 4

**Summary:**

This paper presents EQUAL, a new framework for efficiently extracting instruction-tuning data (QA pairs) from large web-scale corpora. The core idea is to avoid the "extract-all" approach, which is prohibitively expensive. Instead, EQUAL first uses a "warm-up" phase (sampling 5% of the data) to train a contrastive learning model that aligns document embeddings with the QA pairs they contain. It then clusters the entire corpus and uses a Multi-Armed Bandit strategy to iteratively sample from the most promising clusters. The "reward" for the MAB is based on the Optimal Transport distance to a small, trusted reference set. The authors show this method significantly cuts costs (5-10x) while also improving downstream model performance on math, code, and open-ended tasks.

**Strengths:**

1. The paper applied an innovative application of the Multi-Armed Bandit framework to sample from the most promising clusters iteratively.
2. The paper's key insight is interesting: don't cluster raw documents, cluster them by the data you can get out of them.
3. The paper is easy to follow and authors have done quite a lot of experiments including comprehensive downstream benchmarks. My core issues with this paper are that its impressive results seem to rest on a few critical assumptions that don't hold up to scrutiny, making me question its real-world practicality.

**Weaknesses:**

My core issues with this paper are that its impressive results seem to rest on a few critical assumptions that don't hold up to scrutiny, making me question its real-world practicality.
1. The entire framework is anchored to the Optimal Transport distance to a reference set, D_r. The paper's claim (in Appendix P) that using just 20 reference samples is nearly as effective as 1500 is not credible to me. Estimating a stable OT distance for high-dimensional embeddings from 20 samples is statistically dubious. More importantly, this setup assumes the reference set is a perfect, unbiased oracle. In reality, a small D_r is likely biased. The current framework doesn't just use this bias; it optimizes for it.
2. The "cold start" cost is still prohibitive. The paper frames this as a low-cost solution, but the "warm-up" requires processing 5% of the corpus with a 72B-parameter model. For their 1.4M document dataset, that is 70,000 documents. This is not a "warm-up"; it's a massive, expensive inference job in itself. This barrier to entry is extremely high and is not adequately acknowledged.
3. There might be a risk of cascading bias. Any bias in the reference set D_r gets learned into the contrastive learner, which in turn skews the entire clustering. Since the MAB only explores at the cluster level, it has no way to find high-quality, novel documents if they were already buried in a "bad" cluster. The method risks locking onto a local optimum very early in the process.

**Questions:**

1. Can you run an experiment that proves this isn't just a "bias amplifier"? For instance, what happens if you use a D_r for the math task that only contains simple, 1-step arithmetic? Does EQUAL (as I suspect) fail to select clusters with complex, multi-step proofs?
2. The 5% warm-up cost is a major barrier. How does the entire pipeline's performance (end-to-end) degrade if you only use a 1% or 0.1% warm-up budget? What is the minimum compute needed for this to actually work?
3. How much of this success is just an artifact of using a giant 72B model for extraction? What happens if I use a smaller, less-capable 7B model? Does EQUAL still find the best data, or does it just find the data that is easiest for the weak extractor to parse?

---

> ### Author Response · Authors · 2025-11-21
>
> $\textbf{Overall Response:}$
>
> Many thanks for your insightful feedback! We have added 4 additional experiments and provided sufficient discussion w.r.t. your concerns about real-world practicality, warm-up budget and the extractor model, etc. We have also included these experiments and discussion into the revised paper and promise to include them into the final version if the paper is accepted.
>
> $\textbf{Response to Weakness 1:}$
>
> Thanks for your valuable comments! We would like to clarify that the 20 reference samples do not have much bias because
> the evaluation dataset (`GSM8K`, a grade-school word problem with relatively simple reasoning) is relatively simple and homogeneous. Hence, using the 20 samples achieved performance comparable to 1500. For example, `LESS` [1], `NICE` [2], `GREATS` [3] also use several samples (5-20) as the reference set for data selection under such a homogeneous setting.
> To further examine this issue based on the reviewer's comment, we extend our analysis to more complex and diverse benchmarks such as `MATH`} (competition-level problems with long, multi-step proofs). The results show that, for the more challenging dataset, the OT-based distance estimation indeed requires more reference samples (150 samples) to reach stable and effective performance. We have added these additional experiments and discussions to the revised version (Appendix P).
>
> | #-Reference data | 20     | 50     | 75     | 150    | 500    | 1000   | 1500   |
> |-----------------|--------|--------|--------|--------|--------|--------|--------|
> | Accuracy        | 34.37% | 35.10% | 35.76% | 36.11% | 36.19% | 36.23% | 36.40% |
>
>
> [1] Selecting Influential Data for Targeted Instruction Tuning
>
> [2] NICE: Data Selection for Instruction Tuning in LLMs with Non-differentiable Evaluation Metric
>
> [3] GREATS: Online Selection of High-Quality Data for LLM Training in Every Iteration
>
> $\textbf{Response to Weakness 2:}$
>
> Nice Comment! Following the advice of the reviewer, we have further analyzed the impact of the warm-up ratio, varying it from 0.1\%, 1\%, to 5\% of the candidate document pool. Our new experiments confirm that while using 5\% of the corpus indeed incurs significant computational cost, a 1\% warm-up ratio achieves comparable performance across different scenarios.
>
> | #-Warmup Ratio | 0.1%  | 0.5%  | 1%    | 3%    | 5%    |
> |---------------|-------|-------|-------|-------|-------|
> | Accuracy      | 69.66%| 71.87%| 72.91%| 72.96%| 73.03%|
>
> As shown in the following table, with a 1\% warm-up ratio, the warm-up stage accounts for only a small portion (approximately 10\%) of the overall computational cost in our pipeline. Accordingly, we have updated the paper to emphasize that the proposed method remains effective with a 1\% warm-up ratio, which greatly reduces the computational overhead and lowers the entry barrier for practical use. This modification aligns with the reviewer’s concern and better supports our claim of a low-cost, scalable data selection framework.
>
> | Step                                         | FLOPs  |
> |----------------------------------------------|-------:|
> | Clustering                                   | 2.31   |
> | Extracting 1% QA pairs for warm-up           | 1.31   |
> | Fine-tuning the contrastive learning model   | 0.21   |
> | Document selection                           | 0.06   |
> | Extracting 5% QA pairs  for fine-tuning                   | 6.94   |
> | Fine-tuning the LLM on 5% QA pairs           | 1.89   |
> | **Total**                                    | **12.72** |
>
>
> We have included the above discussion into Section 4.3 of the revised paper.
>
> $\textbf{Response to Weakness 3:}$
>
> Thanks for the comment! We would like to clarify that in our paper, the clustering step is conducted without relying on $D_r$.
> To be specific, we introduce the warm-up step in Section 3.2, which aligns the embedding spaces of documents and their extracted QA pairs via contrastive learning on a small, randomly sampled subset of the candidate document pool $D_c$. This step merely encourages document embeddings to be more consistent with the embeddings of the QA pairs they generate, thereby improving clustering quality within $D_c$ itself, without relying on $D_r$.
>
> Furthermore, the reference set $D_r$ is introduced in the clustering-based sampling stage (Section 3.3), where it is used to assess how each candidate cluster contributes to model improvement. To avoid the local optimum, we adopt a Multi-Armed Bandit (MAB) strategy that balances exploration and exploitation at the cluster level: each cluster is treated as an ``arm'', and the algorithm explores under-sampled clusters to discover new, diverse signals while exploiting high-performing clusters to further refine the model. The uncertainty term in the Upper Confidence Bound (UCB) update drives ongoing exploration, reducing the risk of early convergence to a local optimum.

---

> ### Author Response · Authors · 2025-11-21
>
> $\textbf{Response to Question 1:}$
>
> Many thanks for your insightful question! First, we would like to clarify that we do not incorporate much bias in the homogeneous dataset; and for complicated datasets, we can avoid the bias by increasing the number of instances in the reference set to several hundreds for complicated datasets, as discussed in the response to Weakness 1.
>
> Furthermore, we conduct an additional experiment according to the two scenarios that the reviewer suggested, using two standard math benchmarks: `GSM8K` (grade-school word problems with relatively simple reasoning) and `MATH` (competition-level problems with long, multi-step proofs). Concretely, we use `GSM8K` as the reference set to guide `EQUAL`’s cluster selection. Manual inspection shows that the selected data are dominated by simple instances, as the reviewer suspected.
>
> However, we would like to clarify that if the user aims to improve the model’s ability in multi-step proof, it is not difficult to incorporate multi-step proof data into the reference set $D_r$.
>
> $\textbf{Response to Question 2:}$
>
> Thank you for the valuable suggestion to improve our paper! We agree that a 5\% warm-up budget appears large, and we have further conducted a more detailed analysis of this issue; the full discussion can be found in our response to Weakness 2. We have also include the above discussion into Section 4.3 of the revised paper.
>
> $\textbf{Response to Question 3:}$
>
> Many thanks for this insightful question! We agree that the capability of the extractor model can influence the quality of extracted QA pairs, which may further affect the final performance. Nevertheless, the main strength of our approach lies in our ability to mine high-quality QA pairs from vast amounts of web data, instead of the capacity of the extractors.
>
> Specifically, as shown in the table below, when we use smaller models (i.e., Qwen2.5-7B) for QA pair extraction, the overall absolute performance slightly decreases compared to the Qwen2.5-72B extractor due to weaker linguistic and reasoning capabilities, but the relative improvement trend introduced by `EQUAL` is consistent. That is, `EQUAL` continues to identify high-quality and diverse subsets that yield comparable performance to extraction using the full dataset, while requiring significantly less computational cost.
>
> This result indicates that `EQUAL`’s advantage is not tied to the scale of the extractor model, but instead comes from its capacity to deeply explore data diversity and informational richness. By extracting information from a broader range of web sources, we obtain a more diverse set of QA pairs, rather than merely selecting instances that are easier for the model to parse. Therefore, regardless of whether a strong (i.e., Qwen2.5-72B) or weak extractor (i.e., Qwen2.5-7B) is used, `EQUAL` consistently finds data that leads to model quality comparable to using full-data extraction, with substantially reduced resource consumption.
>
> | Methods                 | GSM8K  | MATH  |
> |------------------------|--------|-------|
> | `Random-Qwen2.5-72B`   | 70.05% | 36.18% |
> | `Full-Qwen2.5-72B`     | 70.28% | 40.02% |
> | `EQUAL-Qwen2.5-72B`    | 74.40% | 41.40% |
> | `Random-Qwen2.5-7B`    | 69.38% | 36.19% |
> | `Full-Qwen2.5-7B`      | 69.72% | 39.16% |
> | `EQUAL-Qwen2.5-7B`     | 73.13% | 40.25% |
>
>
> We have included the above discussion into Section 4.3 of the revised paper.

---

> > ### Author Response · Authors · 2025-11-26
> > **Kindly Request to Reviewer 5USS**
> >
> > Dear reviewer 5USS,
> >
> > Wishing you a happy and blessed Thanksgiving!
> >
> > We would like to express our gratitude to your valuable feedback. We have carefully considered all suggestions and updated our submission accordingly.
> >
> > If you have any question for our paper, please feel free to point out and we will try to address it as soon as possible. We would like to express our sincere gratitude for your valuable comments again. Thanks and looking forward to your reply!
> >
> > Best wishes!
> >
> > Authors

---

### Official Review · Reviewer_kjNN · 2025-10-31

**Soundness:** 2
**Presentation:** 3
**Contribution:** 3
**Rating:** 4
**Confidence:** 3

**Summary:**

This paper aims to explore a scalable and cost-effective approach for automatically extracting high-quality instruction data from large-scale document corpora. To this end, the authors propose the EQUAL framework, which integrates contrastive learning–based embedding alignment with a multi-armed bandit–driven adaptive sampling strategy. By iteratively sampling documents and updating distributional similarity based on optimal transport, EQUAL effectively identifies and extracts high-quality instruction data. Experimental results demonstrate that EQUAL significantly reduces computational costs and outperforms multiple strong baselines across various tasks.

**Strengths:**

1. The paper presents a clear research motivation, and the proposed EQUAL method achieves a favorable trade-off between cost and performance, providing valuable insights for future work.
2. The paper is well-organized, and the experiments in this paper cover three major datasets and multiple tasks, providing comprehensive evidence for the effectiveness of EQUAL.
3. The authors conducted comprehensive ablation studies on the proposed framework and provided thorough analyses of the extracted data, further enhancing the persuasiveness of the proposed EQUAL framework.

**Weaknesses:**

1. In lines 144–148, the authors emphasize that the main goal of this paper is to reduce the number of documents to be extracted in order to lower data construction costs while maintaining high model performance. However, the paper does not clearly show the performance gap between the instruction data extracted under high-cost settings and that obtained using the proposed method. This could be an important issue, as a large gap would call into question the practical significance of the proposed cost optimization.
2. The main experiments do not report the amount of data used, making it difficult to assess the specific impact of data scale on the experimental results. The superior performance of the proposed method might be partially attributed to training on a larger amount of data.

**Questions:**

1. Why did the authors not conduct experiments on the Qwen model and instead choose the relatively outdated Mistral-7B model? Including experiments on a more advanced model could further strengthen the persuasiveness of the paper.
1. How large is the performance gap between EQUAL and high-cost instruction data extraction methods, such as directly using closed-source model APIs? If this gap is small, the proposed method would represent a highly valuable contribution by maintaining strong performance while significantly reducing data extraction costs.
1. In the limitations section, the authors briefly note that EQUAL “may still contain imbalanced or culturally skewed content.” However, it remains unclear what proportion of the final dataset consists of such biased content and how it may affect model performance. It is recommended that the authors provide a brief analysis  of these biases.
1. In the anonymized code repository provided by the authors, the following statement appears: “This repository contains the code for our paper [EQUAL: Efficient Scalable Data Extraction Framework for Instruction Tuning] published at ICML 2025.” Even if the paper was not actually accepted by ICML 2025, such a statement could serve as a unique identifier and potentially reveal the authors’ identities. I am not sure whether this violates the double-blind review policy of ICLR 2026, and I would like the authors to clarify this.

---

> ### Author Response · Authors · 2025-11-21
>
> $\textbf{Overall Response:}$
>
> Many thanks for your insightful feedback! We have added 2 additional experiments and provided sufficient discussion w.r.t. your concerns about high-cost settings, impact of data scale and backbone model choice, etc. We have also included these experiments and discussion into the revised paper and promise to include them into the final version if the paper is accepted.
>
> $\textbf{Response to Weakness 1:}$
>
> Thanks for the comment! If the   ``high-cost setting'' refers to the scenario of extracting QA pairs from all documents, we would like to clarify that the comparison between the high-cost (i.e., `All`, extracting QA pairs from all documents) and low-cost (i.e., `EQUAL`, extracting QA pairs from a fraction (5\%) of all documents) data extraction settings has already been shown in Table 2 of our submission. We observe that for most tasks the extraction of only 5\% of QA pairs produces superior results compared to `All`, demonstrating that `EQUAL` achieves high model performance with significantly reduced data construction overhead. Even for the difficult task MATH, extracting QA pairs from only 20\% of the documents in $\mathcal{D}_c$ can achieve a performance comparable to `All` across all training settings. This is because not all QA pairs extracted from all documents in the candidate document pool might contribute to the targeted capability. To improve clarity, we explicitly highlight the performance comparison between high-cost and low-cost settings in the revised manuscript (Line 373).
>
> On the other hand, if the  ``high-cost setting''  refers to the comparison with closed-source API, we conduct additional experiments as follows: we construct additional baselines (i.e., `Random-closed-source` and `All-closed-source`) by using a closed-source API (e.g., Qwen-Flash) to extract QA pairs under the same settings as in our original paper. Furthermore, we consider two instantiations of `EQUAL`: one where QA pairs are extracted using the open-source model Qwen2.5-72B (i.e., `EQUAL-open-source`), and another where they are extracted using the closed-source Qwen-Flash API (i.e., `EQUAL-closed-source`). As shown in the table below, `EQUAL-open-source` achieves comparable performance on both the GSM8K and MATH benchmarks while incurring zero additional API cost, demonstrating its efficiency. Meanwhile, when using the `EQUAL-closed-source`, we achieve the highest accuracy compared to other methods, while incurring significantly lower cost compared to other methods. This shows that `EQUAL` effectively retains high-quality instruction data while substantially saving API cost.
>
> | Methods                   | GSM8K  | MATH  |
> |--------------------------|--------|-------|
> | `Random-closed-source`   | 70.30% | 31.33% |
> | `All-closed-source`      | 72.23% | 35.02% |
> | `EQUAL-open-source`      | 73.01% | 35.10% |
> | `EQUAL-closed-source`    | 74.61% | 37.13% |
>
> We have included the above discussion into Section 4.4 of the revised paper.
>
> $\textbf{Response to Weakness 2:} $
>
> Thanks for the comment! We would like to clarify that the amount of data used in our experiments was reported in the paper. In lines 298 and 349, we respectively reported the total number of documents and the proportion (i.e. 5\%) used for all baseline methods and our proposed approach. In line 372, we conducted ablation studies for different proportions.
> Therefore, the superior performance of our method is not attributed to training on a larger data scale, but rather to the effectiveness of our data selection strategy.
>
> To improve clarity, we explicitly report  the number of instances used for training in the revised manuscript (Lines 351-352).
>
> $\textbf{Response to Question 1:}$
>
> Thanks for your valuable suggestion to improve our paper! Following your advice, we have conducted additional experiments using the Qwen2.5-7B model. The results in the table below show that our proposed method consistently improves performance on Qwen2.5-7B, demonstrating its strong generalizability across different model architectures.
>
> | Methods       | GSM8K | MATH | FLOPs | HUMANEVAL | MBPP | FLOPs |
> |-------------------|------:|-----:|------:|----------:|-----:|------:|
> | `Random`  | 86.1% | 59.6% | 8.51  | 66.7%     | 75.2% | 6.73  |
> | `Avg-sim`      | 87.6% | 65.2% | 111.73| 68.7%     | 75.9% | 64.15 |
> | `Perplexity`    | 86.1% | 66.7% | 146.52| 69.1%     | 76.1% | 72.22 |
> | `Influence`     | 85.6% | 61.0% | 251.28| 66.9%     | 75.3% | 131.65|
> | `LLM-scoring`   | 86.9% | 67.1% | 17.61 | 69.6%     | 76.4% | 137.96|
> | `Rewriting`    | 87.1% | 61.3% | 19.06 | 67.2%     | 75.2% | 14.56 |
> | `Perplexity-MAB`  | 86.4% | 67.0% | 18.23 | 69.7%     | 76.6% | 14.19 |
> | `Influence-MAB`   | 85.9% | 61.6% | 18.19 | 67.8%     | 75.5% | 14.72 |
> | `EQUAL`           | 89.6% | 71.3% | 17.76 | 73.3%     | 78.0% | 13.30 |
>
> These results confirm the persuasiveness of our approach, and we promise to include these new experiments and analysis in the revised version.

---

> ### Author Response · Authors · 2025-11-21
>
> $\textbf{Response to Question 2:}$
>
> Many thanks for your valuable suggestion to improve our paper! We have conducted further experiments to show the performance gap between `EQUAL` and the high-cost scenario. The full discussion can be found in our response to Weakness 1. We have also included the above discussion into Section 4.4 of the revised paper.
>
> $\textbf{Response to Question 3:}$
>
> We appreciate the reviewer’s thoughtful comment! Given the vast scale of web-crawled data, in reality it is inevitable that some documents may contain content that is inconsistent with human values (e.g., promoting anti-human or extremist ideologies). However, we do not observe such cases in the candidate document pool used in this paper, which we believe is cleaned by the curators. Moreover, since the focus of our research lies in data selection for targeted capabilities improvement, rather than detecting or filtering data with inappropriate or biased content, the mechanisms to identify or remove such content fall beyond the scope of this work. Nevertheless, we fully acknowledge the importance of ethical data usage. We therefore recommend that practitioners using our method ensure that their candidate data pool has been screened for harmful or culturally biased content (e.g., using bias-detection models such as `bert-hateXplain`), in order to prevent potential misuse of our approach. We have incorporated this discussion into the Limitations section of the revised version.
>
> $\textbf{Response to Question 4:}$
>
> Thank you for bringing this to our attention. We sincerely apologize for this oversight. The statement ``This repository contains the code for our paper [EQUAL: Efficient Scalable Data Extraction Framework for Instruction Tuning] published at ICML 2025'' was an unintentional placeholder used during internal testing and was not meant to indicate an actual publication or reveal the authors’ identities.
>
> We confirm that this paper has not been published or accepted by ICML 2025, and the repository was anonymized following the double-blind review policy of ICLR 2026. To avoid any possible confusion, we have removed the statement from the repository and ensured that no identifying information remains.
>
> We apologize for any misunderstanding and appreciate the reviewer’s careful attention to maintaining the integrity of the review process.

---

> ### Author Response · Authors · 2025-11-26
> **Kindly Request to Reviewer kjNN**
>
> Dear reviewer kjNN,
>
> Wishing you a happy and blessed Thanksgiving!
>
> We would like to express our gratitude to your valuable feedback. We have carefully considered all suggestions and updated our submission accordingly.
>
> If you have any question for our paper, please feel free to point out and we will try to address it as soon as possible. We would like to express our sincere gratitude for your valuable comments again. Thanks and looking forward to your reply!
>
> Best wishes!
>
> Authors

---

### Comment · Area_Chair_nwcQ · 2025-11-27
**Reviewer Reminder: Author Rebuttals Available**

Dear Reviewers,

The authors have posted their rebuttals to your reviews.

Please read the authors' responses, assess whether your concerns have been addressed, and update your rating and confidence accordingly.

Your prompt attention to the rebuttals is appreciated.

Best,
AC

---

### Author Response · Authors · 2025-11-28
**General Response to Area Chair**

Dear ICLR 2026 Area Chair,

Wishing you a happy and blessed Thanksgiving!

We would like to thank all the reviewers (`kjNN`, `5USS`, and `wiHJ`) for their thoughtful suggestions on our paper, and appreciate that the reviewers have multiple positive impressions of our work, including:

* **Clear research motivation** (Reviewers `kjNN`, `wiHJ`) and **well-organized writing** (Reviewers `kjNN`, `5USS`), presenting a solution effectively tailored to address existing limitations (Reviewer `wiHJ`).
* **Innovative methodology** (Reviewers `5USS`, `wiHJ`), featuring a reasonable Multi-Armed Bandit framework (Reviewers `5USS`, `wiHJ`) and an interesting insight on clustering based on extractable data rather than raw documents (Reviewer `5USS`).
* **Comprehensive empirical validation** (Reviewers `kjNN`, `5USS`, `wiHJ`), covering three major datasets and multiple tasks (Reviewers `kjNN`, `5USS`) with thorough ablation studies (Reviewers `kjNN`, `wiHJ`).
* **Favorable cost-performance trade-off** (Reviewer `kjNN`) and **general usefulness** (Reviewer `wiHJ`), providing valuable insights for future work (Reviewer `kjNN`).

> **Note:** We provide a summary of our responses in the following, with most corresponding discussions included in the revised submission, highlighted in blue color for clarity. Specifically, in response to the reviewers' constructive comments, we have updated the Introduction (Section 1) and the ablation studies (Section 4.3), added a new Section 4.4, and expanded the Appendix to include Sections B, P, S, T, and U. For detailed responses, please refer to the feedback for each comment/question point-by-point.

---

### Introduction
We revise the second paragraph of the Introduction to more clearly distinguish our method from existing approaches, specifically highlighting how our framework overcomes the diversity and scalability limitations inherent in current data synthesis and extraction methods. (Reviewer `wiHJ`).

### Ablation Study (Section 4.3)
* We analyze the impact of the warm-up ratio (varying from 0.1% to 5%) on model accuracy, confirming that a **1% ratio** achieves comparable performance to 5% with lower computational costs (Reviewer `5USS`).
* We evaluate different extractor models, replacing the **Qwen2.5-72B** extractor with **Qwen2.5-7B** and **Qwen-Flash**, demonstrating that $\texttt{EQUAL}$'s performance improvements remain consistent regardless of the extractor's size (Reviewer `kjNN` & `5USS`).

### Further Experiments (Section 4.4)
* We conduct additional **Chain-of-Thought (CoT)** data generation experiments, comparing $\texttt{EQUAL}$ against state-of-the-art methods designed for reasoning models like **OpenThoughts** and **s1k**. Results show $\texttt{EQUAL}$ achieves the best performance with the lowest computational cost (Reviewer `wiHJ`).
* We include experiments using Qwen Model (**Qwen2.5-Math-7B-Instruct** and **Qwen2.5-Code-7B-Instruct**) on benchmarks like **AIME24/25** and **LiveCodeBench**, highlighting the method's robustness across different model families (Reviewer `wiHJ`).

### Appendix Updates
* **Appendix B (Limitations):** We expand the discussion on limitations to include a more in-depth analysis of potential biases in web-crawled corpora and ethical data usage (Reviewer `kjNN`).
* **Appendix P (Reference Data Size):** We analyze the impact of reference dataset size, demonstrating that small reference sets suffice for homogeneous tasks, whereas complex reasoning benchmarks necessitate larger sets for robust estimation (Reviewer `5USS`).
* **Appendix S & T (Detailed Experiments):** We provide full experimental details and results for the Qwen model and the CoT data generation scenarios (Reviewers `wiHJ` & `kjNN`).
* **Appendix U (Key Differences):** We add a section clarifying the Key difference Between $\texttt{EQUAL}$ and existing methods, explaining how our method avoids the limited diversity often seen in seed-based synthesis by selecting naturally occurring QA pairs from heterogeneous corpora (Reviewer `wiHJ`).

---

We once again thank the reviewers and the Area Chair for their time and effort, and we appreciate the opportunity to address these concerns thoroughly during the rebuttal.

Best regards,

The authors of Paper 14969

---

### Meta-Review · Area_Chair_XskH · 2025-12-16

**Summary:**

This paper addresses the cost and effectiveness challenges of instruction tuning by proposing EQUAL, which is an efficient and scalable framework for extracting high-quality instruction data from large web data. Specifically, QUAL combines contrastive embedding–based document clustering with a multi-armed bandit strategy to iteratively identify document clusters that yield the most beneficial QA pairs. This design reduces extraction cost while avoiding low-value or harmful instructions. Extensive experiments across multiple benchmarks and tasks confirm the effectiveness of EQUAL.

It received comments from three reviewers. Before the rebuttal, two of them were somewhat negative about the initial submission. One reviewer was positive. The main concerns are reflected in (1) the included models in experiments are not enough; (2) the computational costs introduced by the proposed method could be huge; (3) A series of confusions regarding the model design detail and the experimental detail. The concerns are considered to be addressed by carefully checking the paper and the rebuttal content. Therefore, the AC recommends acceptance.

**Reviewer Concerns:**

The concerns of the three peer reviews were similar in many ways. Reviewers raised questions about the experiments, method issues, and unconvincing/unclear descriptions. A detailed rebuttal was provided subsequently. Reviewers did not respond to the authors. However, the AC checks the paper, questions, and answers, and thinks that the mentioned concerns have been addressed by the rebuttal.

**Reviewer Scores:**

**Reviewer kjNN**
The concerns include (1)  the performance gap between the instruction data extracted under high-cost settings and that obtained using the proposed method; (2) the amount of data used; (3) experiments on the Qwen model; (4) confusing limitation description. The rebuttal includes detailed answers to the concerns. Therefore, if the reviewer had been able to participate fully in the discussion, the score would be positive.

**Reviewer 5USS**
 The concerns include (1) warm-up budget; (2) the used models for extraction; (3) the bias caused by the proposed method. (1) and (2) were addressed by detailed supplemented results. For (3), some results and feasible suggestions were provided. Therefore, the score of this reviewer would be improved based on the discussion.

**Reviewer wiHJ**
The concerns include (1) the diversity problem with the proposed method; (2) the models used; (3) the tasks and datasets. For (1), a supplemented explanation of the high-level idea is provided. For (2), more experiments are supplemented. For (3), the tasks and datasets are supplemented based on the suggestions of the reviewer. The review was positive about the initial submission. During the rebuttal, if he/she can participate fully in the discussion, he/she would keep the support or increase the score to 8.

---

### Decision · Program_Chairs · 2026-01-26

Accept (Poster)